# How the motor system copes with aging: a quantitative meta-analysis of the effect of aging on motor function control

Laura Zapparoli [1,2 ✉], Marika Mariano[1] & Eraldo Paulesu[1,2]

Motor cognitive functions and their neurophysiology evolve and degrade along the lifespan in a dramatic fashion. Current models of how the brain adapts to aging remain inspired primarily by studies on memory or language processes. Yet, aging is strongly associated with reduced motor independence and the associated degraded interaction with the environment: accordingly, any neurocognitive model of aging not considering the motor system is, ipso facto, incomplete. Here we present a meta-analysis of forty functional brain-imaging studies to address aging effects on motor control. Our results indicate that motor control is associated with aging-related changes in brain activity, involving not only motoric brain regions but also posterior areas such as the occipito-temporal cortex. Notably, some of these differences depend on the specific nature of the motor task and the level of performance achieved by the participants. These findings support neurocognitive models of aging that make fewer anatomical assumptions while also considering tasks-dependent and performance-dependent manifestations. Besides the theoretical implications, the present data also provide additional information for the motor rehabilitation domain, indicating that motor control is a more complex phenomenon than previously understood, to which separate cognitive operations can contribute and decrease in different ways with aging.

[1] Psychology Department and NeuroMi – Milan Centre for Neuroscience, University of Milano-Bicocca, Milan, Italy. [2] IRCCS Orthopedic Institute Galeazzi, Milan, Italy. ✉email: laura.zapparoli@unimib.it

People worldwide live longer: for the first time in history, most people can expect to live into their sixties and beyond. Based on the World Health Organization projections, the world's population aged 60 years and beyond is expected to total 2 billion by 2050, more than twice as many in 2015[1]. Cognitive functioning is of great importance in older age. Preserving cognitive functioning delays care dependency and reverse physical frailty; it follows that understanding age-related cognitive decline and its neural mechanisms is a major topic of research in the domain of cognitive neuroscience. The advent and availability of magnetic resonance functional imaging methods have significantly advanced the understanding of how aging affects the neurobiological functions at multiple levels leading to changes in brain function[2–5].

It is widely known that even healthy aging is associated with grey matter volume reductions and functional brain changes, together with cognitive decline in a variety of domains:[6–11] memory[12–15], sensory perception[16–19], attention and executive functions[20–22], and language[23–25].

However, only a limited number of studies have focused on the effects of physiological aging on motor control, which we define here as a gradual decrease of motor abilities associated with aging in the absence of a significant breakdown of specific neural systems nor accompanying symptoms like, for example, tremor, rigidity, or typical gait disorders. This is quite surprising since the quality of life of older adults is strongly associated with their motor independence and the subsequent efficient interactions with the environment.

In the present paper, we address this specific topic by reviewing with a quantitative meta-analysis the available evidence derived from task-based brain imaging activation studies on how the physiological process of aging affects motor control. Crucially, we considered not only explicit motor execution tasks, but also cognitive-motor tasks (e.g., motor imagery, motor prediction paradigms, typically used to investigate motor functions, see for example refs. 26,27) and the achieved level of behavioural performance: this last aspect was taken as an indirect index of the task demands. To the best of our knowledge, this is the first meta-analytical attempt to summarize the previous findings on aging in the domain of motor control with this specific approach.

Before presenting our study, we first introduce the current neurocognitive theories on how healthy aging affects brain functioning. We then summarize the evidence derived from previous reviews and meta-analyses conducted on this topic. Finally, we spell out the methodological features of the present study that make it novel in comparison with previous meta-analyses of aging and the motor system.

There is a rich body of literature documenting age-related patterns of brain activity associated with the changes in cognitive functions that are typical of physiological aging[28], interpreted by several cognitive models.

On the one hand, the compensation hypothesis predicts that age-related increases in brain activation, as well as the recruitment of additional areas, compensate for various neural/behavioural deficits[15,29–32]. According to the compensation hypothesis, these overactivations would be larger in good than in poor performers. Compensatory patterns have been documented for several cognitive domains (e.g., working memory, episodic memory retrieval, perception, inhibitory control[15,29–32]). This phenomenon was initially observed in the prefrontal cortex. In particular, compensatory processes have been described as reduced inter-hemispheric asymmetries for tasks associated with strongly lateralized fMRI patterns in younger participants[29]. This overall pattern is well captured by the Hemispheric Asymmetry Reduction in Older Adults (HAROLD model,[30]).

A similar compensatory hypothesis is covered by the Posterior-Anterior Shift in Aging model (PASA,[33]), whereby aging is associated with a significant increment of the frontal lobes' activation and reduced neural activity in posterior brain areas (mainly occipito-temporal cortices). Age-related reductions in occipital activity have been attributed to deficits in sensory processing, while age-related increased prefrontal activity would represent an attempt to compensate for these deficits.

On the other hand, the dedifferentiation hypothesis posits that age-related brain functional changes might indicate a generalized diffusion of brain activity attributable to deficits in neuro-transmission, which in turn causes a loss of neural specialization. This hypothesis suggests that age-related hyperactivations should not be accompanied by successful behavioural performance[34]. Several experiments have previously supported the dedifferentiation hypothesis. For example, Loibl et al.[35] observed age-related increased activations in ipsilateral motor areas (M1 and SMA), that were negatively correlated with motor performance[35]. Similar results were found by Bernard et al.[34] with transcranial magnetic stimulation, reporting for the elderly subjects more diffuse motor cortical excitability in the hemisphere contralateral and ipsilateral to the recorded motor evoked potentials. Notably, this broader excitability correlated with augmented reaction times[34].

However, these hypotheses seem insufficient to capture the dynamics of compensation in elderly subjects when considering the specific task demands. These may change in an age-dependent manner as summarized by the Compensation-Related Utilization of Neural Circuit Hypothesis (CRUNCH hypothesis,[36]) and the posited group by performance interactions. Indeed, this model calls into play the level of task demands, proposing that in tasks characterized by lower task demands, older adults recruit greater neural resources to compensate and reach the same level of behavioural performance as young people. When the task demands increase, they show equivalent or lower activation and worse behavioural performance compared with young adults[36]. Authors describe this phenomenon as a tradeoff phenomenon that may explain why, as the task demand increases, the younger individuals can reach a good level of performance compared with elderly people who show a declined performance[36].

More recently, the same authors proposed the so-called Scaffolding Theory of Aging and Cognition model[28], defining how and why the compensation through overactivation arises in the aging brain. In the same model, they introduce the concept of scaffolding to describe neural adaptations associated with the life cycle. The basic idea is that the brain is a dynamic organism seeking to maintain homeostatic cognitive functioning. Aging would be associated with a continuous functional reorganization and functional repairs that result in self-generated support to cognitive functioning through the scaffolding. The innovative insight is that scaffolding is not considered a hallmark of the aging brain alone. Instead, it would be a process that characterizes the entire lifespan. In fact, in young individuals, the main aim of the scaffolding process is to respond to challenging situations, e.g., when asked to perform a difficult task or when asked to achieve new skills. However, in elderly people scaffolding processes may be requested even to perform simple tasks due to the degradation of existing neural circuitry that makes basic tasks challenging. This model is particularly important, since it represents the first attempt of linking functional, structural, cognitive changes in aging.

To date, only a few meta-analyses investigated the functional effects of aging on cognitive and neural functioning. Spreng et al.[37] meta-analytically reviewed 77 functional imaging studies tackling different cognitive functions ("memory"[38–40]; "perception";[17,41,42] "executive functions"[43–45]; "motor functions"[46]): they found that old adults activate prefrontal regions more than young adults. Interestingly, when the behavioural performance was similar

between the two groups, older adults engaged more the left prefrontal cortex, while the right prefrontal cortex was more activated when a behavioural decline was present. On the other hand, young adults recruited occipital regions more than older adults, particularly when performance was unequal and during perceptual tasks[37]. More recently, Li et al.[47] described similar results in a meta-analysis conducted on 115 functional imaging studies. The authors found a general decline in the behavioural performance recorded during the fMRI scanning in older adults. This decline was associated with hypoactivations of the visual network and hyperactivations of fronto-parietal and default mode networks. As in Spreng et al.[37], there was a significant relationship between behavioural and neurofunctional findings, since the degree of increased activation in the fronto-parietal network was positively associated with the behavioural performance of older adults[37,47]. The results of both meta-analyses partially support the HAROLD model, since they showed increased activations of the prefrontal cortex in older adults (even if not bilaterally). Moreover, older adults presented significant hypoactivations of the visual network, a finding in line with PASA model—even though such pattern did not replicate for all the cognitive domains explored[47]. Accordingly, it looks as if the HAROLD and the PASA models of aging, with their detailed anatomical predictions, do not capture fully the age-related brain changes described in later studies. Moreover, as for most of the studies on which the HAROLD, PASA, and CRUNCH models were formulated, the experiments included in these previous meta-analyses largely addressed the domains of episodic memory, working memory or reasoning, leaving motor cognition largely neglected.

Aging-related prefrontal hyperactivations are also reported by studies addressing the effects of aging on motor control. Indeed, Seidler et al.[48] reviewed this literature to report that older adults show additional activations in the higher-level prefrontal region, but also in sensorimotor cortical areas. Notably, the activity of these prefrontal and sensorimotor regions was often associated with better performance, suggesting the possibility that the engagement of additional areas may represent a compensatory process taking place during motor task performance[48].

This was partially in line with the results of the only available meta-analysis exploring the effects of aging on the neurofunctional correlates of motor control. Turesky and colleagues[49] conducted an activation likelihood estimation (ALE) meta-analysis on older adults and young adults all performing paced right-hand finger movement tasks. The authors found that elderly subjects activated more frequently the right sensorimotor cortex, the right supramarginal gyrus, the medial premotor cortex, and the right posterior cerebellum. They concluded that older adults showed augmented ipsilateral recruitment of brain regions during right-hand finger movements[49].

However, this paper did not explore the possible interactions of the factor aging with the level of performance achieved by the subjects and with the nature of the motor task, something that we aimed at with the present study, where we also considered cognitive-motor tasks (e.g., motor imagery, action prediction, action observation). Moreover, the mentioned meta-analysis only focused on one specific (explicit) motor task performed with the same limb, leaving unexplored the possibility that such findings might generalize independently from the specific motor task and the effector involved.

The imaging literature, with about 40 specific empirical papers focused on various motor tasks executed by elderly and young participants, is now sufficiently mature to try and fill this gap, and test how good are the neurocognitive models in capturing the available evidence. The obvious tools to try and test the goodness of such data fitting are quantitative meta-analyses, particularly those performed with techniques that permit to test articulated

factorial designs, as the designs implied by the intersections of the factors alluded to earlier on, like age, nature of the task and level of performance/task demands. This is what we describe next with the specific aims of our study.

In this study, we investigated the effects of aging on the neurofunctional correlates of motor control, testing the different neurocognitive theories of aging with a meta-analytical approach. Besides the aging process, we considered two other factors: the specific nature of the tasks (with reference to the presence/absence of an overt motor output: motor execution tasks vs cognitive-motor tasks) and the task demands (inferred by the level of performance achieved by the elderly subjects: similar to young participants vs inferior). We focused on evidence coming from fMRI/PET activation studies on groups of young and elderly individuals. As the reader will see below, we identified neuroimaging studies either on explicit motor execution tasks (i.e., simple motor paradigms such as finger-tapping/opposition tasks, bimanual or face/mouth movements; more complex paradigms such as repeated flexion-extension movements, force modulation tasks, drawing tasks) or cognitive-motor tasks (i.e., motor imagery, motor observation and motor prediction). These tasks were executed mainly by right-handed participants.

Crucially, we decided to include a variety of experimental tasks involving different body districts, executed with the right side and/or the left side or bilaterally. This was done to investigate whether potential aging-related neurofunctional effects might occur regardless of the specific effector and/or the side used to execute the task.

We combined a hierarchical clustering (HC) algorithm with the coordinate-based ALE. The former allowed us to assess the data according to a factorial design, while the ALE method allowed us to assess the spatial significance of the clusters identified with HC. We assessed whether (i) motor control is associated with aging-related meaningful changes in brain activity, (ii) these modifications depend on the level of task demands and/or on the specific nature of the motor task, (iii) how these changes can be interpreted in the light of the neurocognitive models of ageing.

In which brain regions should we expect such effects? One possibility is the involvement of cortical and subcortical areas typically associated with motor control, such as the primary sensorimotor cortices, the supplementary motor area, premotor cortex, basal ganglia, and cerebellum, see[50]. Other possible brain candidates are the regions involved in cognitive-motor tasks, such as fronto-parietal networks activated during motor imagery tasks[51], but also occipito-temporal regions typically recruited during action prediction and/or action observation paradigms[52].

Admittedly we did not commit ourselves to any of the aforementioned neurocognitive models from the outset: indeed, we consider these not to be necessarily mutually exclusive. Yet, we had clear the specific findings that would be in support of any given theory.

On the one hand, the HAROLD model would be confirmed by an overactivation of the prefrontal cortices in elderly participants, specifically in association with a good behavioural performance. Since we consider here motor control functions, one possible extension of this model would imply increased bilateral recruitment of sensorimotor regions (see Turesky et al.[49]).

On the other hand, the PASA model would be satisfied by a general reduction of the activity of the posterior regions (mainly occipito-temporal), together with a significant increment of activations at the level of the frontal lobe, again associated with the performance level.

Finally, the CRUNCH hypothesis would be met by the observation, in older adults of overactivations at the level of task-specifics brain networks for low-demanding tasks and consequent

behavioural performance similar to the young counterparts and hypoactivations in task-specifics brain networks for high-demanding tasks and consequent low-level behavioural performance.

## Results

**Qualitative description of the behavioural data.** Twenty-one out of 40 studies described a behavioural decline in the elderly participants, suggesting a general deterioration of motor functions associated with the physiological process of aging: older adults showed increased reaction times (RTs, 5 studies), or reduced accuracy (12 studies), or both increased RTs and reduced accuracy (4 studies) during the execution of different motor tasks (see Table 1 and Supplementary Data 1).

To assess whether impaired performance in older adults was more frequently associated with the specific nature of the task, we performed a chi-squared analysis (Factor1, performance: Elderly = young/elderly < young, considering both accuracy and RTs data; Factor2, nature of the task: motor execution tasks/cognitive-motor tasks). The chi-squared test of independence showed that there was no significant association between the level of behavioural performance achieved by the elderly participants and the nature of the task, $X^2(1, N = 40) = 1.58, p = 0.21$.

**HC and cluster composition analysis (CCA).** The HC analysis identified 66 clusters. The mean standard deviation along the three axes was 5.86 mm (x-axis), 6.54 mm (y-axis) and 6.90 mm (z-axis). On average these clusters contained 21 foci (range: 8–43). Of these, 22 were retained after the intersection analysis with the ALE map. On average, these overlapping clusters contained 24 foci (range: 10–43). Only these clusters were then submitted to a CCA to test the peaks distribution with reference to our factors of interest (group, performance, and task) and their interactions.

Given the aim of our study of investigating the effects of aging on the neurophysiology of motor control, we will report in the main text only main effects and interactions involving the factor "Group" (i.e., group-specific clusters, Group-by-Performance interaction, Group-by-Task interaction, and Group-by-Performance-by-Task). The other results are described in the Supplementary Note (see also Supplementary Figs. 1–3).

**Group-specific clusters.** The binomial CCA performed on the factor "Group" revealed that two clusters were significantly associated with the elderly group and two clusters were more frequently activated in the young subjects. The elderly-specific clusters were in the left calcarine fissure (CL6) and left superior occipital gyrus (CL9), while young-specific clusters were located in the left supramarginal gyrus (CL11) and in the right sensorimotor regions (precentral gyrus/postcentral gyrus, CL45). See Table 2 and Fig. 1.

The peaks that contributed to such clusters derived from seven different studies (left calcarine fissure, CL6), seven different studies (left superior occipital gyrus, CL9), six different studies (left supramarginal gyrus, CL11) and nineteen different studies (right precentral gyrus/postcentral gyrus, CL45). These peaks are listed in Supplementary Data 2. A further classification of peaks forming the sensorimotor clusters based on the effector/body side is available in Supplementary Data 3. Based on this last classification, we were able to test the effect called "age-related reduction of hemispheric lateralization", whereby older adults might show a greater involvement of sensorimotor regions on the same side of the body involved by the task. We observed that this was true for CL45 (right precentral gyrus/postcentral gyrus). Indeed, the chi-squared analysis (Factor1, Group: Elderly/Young;

Factor2, body side: left/right) showed a trend of significance ($X^2(1, N = 15) = 3.6, p = 0.056$): older adults showed a higher number of peaks derived from tasks performed with the right side of the body (#peaks left body side elderly: 5; #peaks left body side young: 4; #peaks right body side elderly: 6; #peaks right body side young: 0; #peaks bilateral body side elderly: 11; #peaks bilateral body side young: 14).

**Group-by-performance interaction.** We identified three clusters showing a significant group-by-performance interaction effect. These were in the left postcentral gyrus (CL12), the left cerebellum (CL39) and the right occipito-temporal cortex (CL66, see Table 2 and Fig. 2). The peaks that contributed to such clusters derived from twelve different studies (left postcentral gyrus, CL12), nine different studies (left cerebellum, CL39) and eleven different studies (right occipito-temporal cortex, CL66). These peaks are listed in Supplementary Data 2. Moreover, we observed that for CL12 (left postcentral gyrus) older adults showed a similar number of peaks derived from tasks performed with the left- i.e., same- side of the body ($X^2(1, N = 18) = 2.1, p = 0.15$; #peaks left body side elderly: 4; #peaks left body side young: 5; #peaks right body side elderly: 7; #peaks right body side young: 2; #peaks bilateral body side elderly: 3; #peaks bilateral body side young: 3, Supplementary Data 3).

The interaction plots presented in Fig. 2 show that for the right occipito-temporal and for the left sensorimotor cortices there was a more frequent activation in the elderly subjects providing that task performance was balanced across groups (see orange bars). This represents a case of successful compensation. In the case of the left sensorimotor cortex the activation was rather more frequent in young participants when the elderly subjects had a reduced performance (yellow bars), a case of failed compensation. On the other hand, the right cerebellum was more frequently activated by the elderly subjects in case of reduced performance and equally activated by young and elderly subjects in case of similar performance, a case of compensatory attempt.

**Group-by-task interaction.** Only one cluster, located in the right precuneus (CL58), showed a significant group-by-task interaction. The interaction plot presented in Fig. 3 shows that this region is more frequently activated in the elderly during the execution of motor tasks, while in the young group it is recruited only during the cognitive-motor tasks. The peaks that contributed to such cluster derived from eleven different studies, described in Supplementary Data 2.

**Group-by-task-by-performance interaction qualitative exploration.** Two clusters, located in the left sensorimotor cortices (CL12) and in the right occipito-temporal region (CL66), displayed both a significant group-by-performance and performance-by-task interaction effect (Table 2, Fig. 4), suggesting a possible three-way interaction effect.

The inspection of the graph in Fig. 4a shows that the left sensorimotor region was more likely to be associated with the elderly group during the execution of motor execution tasks, when their performance is similar to their young counterparts and with the young participants group during the execution of cognitive-motor tasks, when they achieved a better behavioural performance.

The inspection of the graph in Fig. 4b shows that this region was more likely to be associated with the elderly group during the execution of cognitive-motor tasks, when their performance is similar to their young counterparts. Interestingly, this cluster falls within the boundaries of a right extra-striate body area[53] and its

**Table 1 Neuroimaging studies included in the current meta-analysis.**

| | First Author | Year | Technique | Sample size (E/Y) | Female (E/Y) | Mean age (E/Y) | Task | Motor district | Side | Participant's handedness | Level of performance |
|---|---|---|---|---|---|---|---|---|---|---|---|
| 1 | Allali | 2014 | fMRI | 14/14 | 10/10 | 66/27 | Cognitive-motor task | Lower | Bilateral | Right | Young > elderly (augmented RTs in elderly subject) |
| 2 | Calautti | 2001 | PET | 7/7 | 4/4 | 60.4/24.4 | Motor execution task | Upper | Bilateral | Right | Young = elderly |
| 3 | Coxon | 2016 | fMRI | 20/20 | 11/11 | 68.7/25 | Motor execution task | Upper | Bilateral | Right | Young > elderly (augmented RTs in elderly subject) |
| 4 | Diersch | 2013 | fMRI | 15/19 | 10/14 | 61.1/22.6 | Cognitive-motor task | Upper and Lower | Bilateral | Right | Young > elderly (reduced accuracy in elderly subjects) |
| 5 | Goble | 2010 | fMRI | 16/16 | 8/8 | 68.3/25.7 | Motor execution task | Upper | Bilateral | Right | Young > elderly (reduced accuracy in elderly subjects) |
| 6 | Godde | 2018 | fMRI | 12/12 | N/A | 57.75/29.67 | Motor execution task | Upper | Right | Right | Young > elderly (reduced accuracy in elderly subjects) |
| 7 | Heuninckx | 2005 | fMRI | 10/11 | 6/6 | 64.8/22.4 | Motor execution task | Upper and Lower | Right | Right | Young = elderly |
| 8 | Heuninckx | 2010 | fMRI | 12/12 | 5/9 | 66.9/23.5 | Motor execution task | Upper and Lower | Right | Right | Young > elderly |
| 9 | Heuninckx | 2008 | fMRI | 26/12 | 12/6 | 65.7/22.4 | Motor execution task | Upper and Lower | Right | Right | Young > elderly (reduced accuracy in elderly subjects) |
| 10 | Hughes | 2010 | fMRI | 15/28 | 8/11 | 66.5/31 | Motor execution task | Upper | Right | Right | Young > elderly (augmented RTs in elderly subjects) |
| 11 | Humbert | 2009 | fMRI | 11/12 | 6/6 | 72.3/27.9 | Motor execution task | Face | Bilateral | Right and Left (3 participants) | Young > elderly |
| 12 | Kim | 2010 | fMRI | 26/20 | N/A | 65.5/23 | Motor execution task | Upper | Right | Right | Young = elderly |
| 13 | Kiyama | 2014 | fMRI | 20/20 | 9/10 | 68.2//25.2 | Motor execution task | Upper | Bilateral | Right | Young > elderly (reduced accuracy in elderly subjects) |
| 14 | Langan | 2010 | fMRI | 18/18 | 9/9 | 71.7/21.4 | Motor execution task | Upper | Right | Right | Young > elderly (augmented RTs and Reduced accuracy in elderly subjects) |
| 15 | Mattay | 2002 | fMRI | 12/10 | 5/1 | 59/30 | Motor execution task | Upper | Right | Right | Young > elderly (augmented RTs and reduced accuracy in elderly subjects) |
| 16 | Michels | 2018 | fMRI | 11/18 | N/A | 62.6/30.3 | Motor execution task | Upper | Bilateral | Right | Young = elderly |
| 17 | Michely | 2018 | fMRI | 12/12 | 0/0 | 62.1/27.4 | Motor execution task | Upper | Bilateral | Right | Young > elderly (free condition) Young = elderly (intern and extern conditions) |
| 18 | Mouthon | 2018 | fMRI | 16/16 | 7/6 | 72/27 | Cognitive-motor task | Lower | Bilateral | N/A | Young > elderly (augmented RTs and reduced accuracy in elderly subjects) |
| 19 | Nedelko | 2010 | fMRI | 13/13 | 6/7 | 63/26.2 | Cognitive-motor task | Upper | N/A | Right | Young = elderly |
| 20 | Onozuka | 2003 | fMRI | 13/11 | 5/4 | 69/22.5 | Motor execution task | Face | Bilateral | N/A | Young > elderly |
| 21 | Papegaaij | 2017 | fMRI | 32/23 | 32/23 | 73.9/23.6 | Motor execution task | Lower | Bilateral | N/A | Young > elderly (reduced accuracy in elderly subjects) |
| 22 | Riecker | 2006 | fMRI | 10/10 | 5/5 | 66/23 | Motor execution task | Upper | Right | Right | Young = elderly |
| 23 | Rodriguez-Aranda | 2020 | fMRI | 17/15 | 8/7 | 70.5/29 | Motor execution task | Upper | Bilateral | Right | Young > elderly (augmented RTs in elderly subject) |
| 24 | Roski | 2014 | fMRI | 20/20 | 11/9 | 65/25 | Motor execution task | Upper | Bilateral | Right | Young = elderly |

**Table 1 (continued)**

| | First Author | Year | Technique | Sample size (E/Y) | Female (E/Y) | Mean age (E/Y) | Task | Motor district | Side | Participant's handedness | Level of performance |
|---|---|---|---|---|---|---|---|---|---|---|---|
| 25 | Sacheli | 2020 | fMRI | 21/21 | 9/11 | 66.33/25.48 | Cognitive-motor task | Lower | Bilateral | Right | Young = elderly |
| 26 | Santos Monteiro | 2017 | fMRI | 18/25 | 11/14 | 68.6/21.5 | Motor execution task | Upper | Bilateral | Right | Young > elderly (reduced accuracy in elderly subjects) |
| 27 | Taniwaki | 2007 | fMRI | 12/12 | 5/3 | 62.9/24.9 | Motor execution task | Upper | Left | Right | Young = elderly |
| 28 | Tremblay | 2017 | fMRI | 14/13 | 10/8 | 68.2/26.8 | Motor execution task | Face | Bilateral | Right | Young > elderly (augmented RTs in elderly subject) |
| 29 | Van Impe | 2009 | fMRI | 21/17 | N/A | 70.3/23.8 | Motor execution task | Upper and Lower | Right | Right | Young > elderly (reduced accuracy in elderly subjects) |
| 30 | Van Impe | 2011 | fMRI | 20/20 | 11/11 | 68/25.2 | Motor execution task | Upper and Lower | Right | Right | Young > elderly (reduced accuracy in elderly subjects) |
| 31 | Wai | 2012 | fMRI | 13/14 | 6/7 | 64.8/21.5 | Cognitive-motor task | Lower | Bilateral | N/A | Young = elderly |
| 32 | Wang | 2014 | fMRI | 20/19 | 12/7 | 62.5/21.6 | Motor execution task + cognitive-motor task | Upper | Bilateral | Right | Young > elderly (reduced accuracy in elderly subjects) |
| 33 | Ward | 2003 | fMRI | 26 | N/A | 21-80 | Motor execution task | Upper | Bilateral | Right | Young = elderly |
| 34 | Ward | 2008 | fMRI | 40 | N/A | 21-75 | Motor execution task | Upper | Bilateral | Right | Young = elderly |
| 35 | Wittenberg | 2014 | fMRI | 12/12 | 5/6 | 66.8/29 | Motor execution task | Upper | Bilateral | Right and ambidextrous (1 participant) | Young = elderly |
| 36 | Wu | 2005 | fMRI | 12/12 | 4/4 | 61.8/30.5 | Motor execution task | Upper | Right | Right | Young = elderly |
| 37 | Zapparoli | 2019 | fMRI | 22/22 | N/A | 61/27.5 | Cognitive-motor task | Upper | Bilateral | Right | Young > elderly (augmented RTs and reduced accuracy in elderly subjects) |
| 38 | Zapparoli | 2016 | fMRI | 29/27 | 14/15 | 61/31 | Cognitive-motor task | Upper | Bilateral | Right | Young = elderly |
| 39 | Zapparoli | 2013 | fMRI | 24/24 | 13/12 | 60/27 | Motor execution task + cognitive-motor task | Upper | Bilateral | Right | Young = elderly |
| 40 | Zwergal | 2012 | fMRI | 60 | 30 | 24-78 | Cognitive-motor task | Lower | Bilateral | N/A | Young = elderly |

For each study, we report the first author, the publication's year, the technique used, the sample size, the type of task, the performance of the two groups. More details are reported in Supplementary Data 1.

**Table 2 Results of the cluster composition analysis.**

| Anatomical label (BA) | Cluster ID | # of Peaks | Hemisphere | Stereotaxic coordinates | | | Group effects | | Performance effects | | Task effects | | Interaction effects | | |
| --- | --- | --- | --- | --- | --- | --- | --- | --- | --- | --- | --- | --- | --- | --- | --- |
| | | | | X (sd) | Y (sd) | Z (sd) | Elderly | Young | Equal (E = Y) | Non equal (E < Y) | Motor execution tasks | Cognitive-motor tasks | Group-by-performance | Group-by-task | Performance-by-task |
| Precentral gyrus (6) | 51 | 19 | R | 44 (5.9) | −6 (5.9) | 52 (4.6) | 0.267 | 0.876 | 0.678 | 0.498 | 0.996 | 0.014* | 0.264 | 1 | 0.393 |
| Postcentral gyrus (4/6) | 65 | 37 | R | 52 (7.9) | 0 (8.5) | 32 (5.6) | 0.788 | 0.326 | 0.989 | 0.025* | 0.643 | 0.494 | 0.311 | 0.165 | 1 |
| Precentral gyrus (4)/Postcentral gyrus (3) | 12 | 24 | L | −53 (4.8) | −6 (6.6) | 35 (8.7) | 0.97 | 0.073 | 0.853 | 0.261 | 0.874 | 0.236 | 0.034* | 0.4 | 0.008* |
| Precentral gyrus (4)/Postcentral gyrus (3) | 27 | 43 | L | −37 (5.0) | −21 (6.6) | 53 (6.4) | 0.783 | 0.327 | 0.976 | 0.047* | 0.002* | 1 | 1 | 0.568 | 0.638 |
| Precentral gyrus (3) | 28 | 28 | L | −29 (8.0) | −7 (7.2) | 53 (5.2) | 0.219 | 0.893 | 0.219 | 0.881 | 0.985 | 0.037* | 0.313 | 0.633 | 0.671 |
| Postcentral gyrus (3) | 29 | 20 | L | −34 (5.7) | −37 (8.5) | 62 (4.1) | 0.568 | 0.633 | 0.507 | 0.672 | 0.01* | 0.999 | 1 | 1 | 0.25 |
| Precentral gyrus (4)/Postcentral gyrus (3) | 45 | 43 | R | 39 (4.9) | −23 (3.9) | 53 (5.5) | 0.991 | 0.019* | 0.124 | 0.929 | 0.049* | 0.978 | 1 | 0.256 | 0.044* |
| Supramarginal gyrus (40) | 11 | 12 | L | −59 (1.8) | −22 (2.7) | 39 (3.9) | 0.996 | 0.018* | 0.729 | 0.49 | 0.017* | 1 | 0.099 | 1 | 1 |
| Inferior Parietal lobule (40) | 30 | 33 | L | −41 (9.1) | −47 (8.8) | 43 (6.7) | 0.334 | 0.8 | 0.778 | 0.34 | 1 | <0.001* | 0.215 | 0.672 | 0.086 |
| Superior Temporal gyrus (42) | 60 | 17 | R | 41 (7.9) | −25 (6.8) | 5 (9.8) | 0.887 | 0.239 | 0.296 | 0.849 | 0.999 | 0.004* | 0.323 | 0.6 | 0.618 |
| Superior Temporal gyrus (42) | 61 | 32 | R | 61 (4.7) | −29 (8.4) | 20 (6.7) | 0.921 | 0.147 | 0.93 | 0.13 | 0.98 | 0.045* | 0.72 | 1 | 0.072 |
| Precuneus (5) | 58 | 14 | R | 13 (6.9) | −53 (6.0) | 61 (7.6) | 0.151 | 0.956 | 0.959 | 0.116 | 0.14 | 0.96 | 1 | 0.02* | 1 |
| Inferior Temporal gyrus (37)/Inferior Occipital Gyrus (19) | 66 | 23 | R | 47 (5.3) | −66 (6.3) | −2 (8.1) | 0.419 | 0.748 | 0.227 | 0.881 | 0.999 | 0.003* | 0.048* | 0.35 | 0.033* |
| Superior Occipital gyrus (19) | 9 | 10 | L | −27 (6.4) | −70 (3.8) | 36 (6.6) | 0.048* | 1 | 0.446 | 0.789 | 0.96 | 0.132 | 1 | 1 | 0.198 |
| Calcarine fissure (17) | 6 | 15 | L | −8 (4.3) | −84 (10.8) | 2 (9.5) | 0.01* | 1 | 0.28 | 0.874 | 0.997 | 0.012* | 1 | 1 | 0.032* |
| Cerebellum 4_5 | 42 | 21 | R | 16 (5.6) | −52 (3.5) | −21 (2.6) | 0.531 | 0.656 | 0.013* | 0.997 | 0.169 | 0.93 | 0.537 | 0.244 | 0.539 |
| Cerebellum 6 | 49 | 16 | R | 7 (6.0) | −81 (6.6) | −12 (6.9) | 0.433 | 0.767 | 0.986 | 0.047* | 0.99 | 0.033* | 0.227 | 1 | 0.588 |
| Cerebellum 6 | 39 | 17 | R | 31 (4.4) | −49 (7.1) | −27 (6.5) | 0.579 | 0.628 | 0.669 | 0.518 | 0.875 | 0.258 | 0.04* | 0.137 | 1 |
| Cerebellum 6 | 4 | 16 | L | −26 (4.5) | −48 (4.5) | −26 (4.9) | 0.964 | 0.099 | 0.222 | 0.906 | 0.032* | 0.996 | 0.598 | 0.424 | 0.171 |
| Vermis 4_5 | 43 | 24 | R | 5 (4.0) | −58 (8.9) | −12 (8.5) | 0.368 | 0.787 | 0.013* | 0.996 | 0.343 | 0.806 | 0.551 | 0.117 | 1 |

**Table 2 (continued)**

| Anatomical label (BA) | Cluster ID | # of Peaks | Hemisphere | Stereotaxic coordinates | | | Group effects | | Performance effects | | Task effects | | Interaction effects | | |
|---|---|---|---|---|---|---|---|---|---|---|---|---|---|---|---|
| | | | | X (sd) | Y (sd) | Z (sd) | Elderly | Young | Equal (E = Y) | Non equal (E < Y) | Motor execution tasks | Cognitive-motor tasks | Group-by-performance | Group-by-task | Performance-by-task |
| Pallidum | 47 | 31 | R | 23 (9.2) | −5 (7.3) | 2 (5.7) | 0.972 | 0.061 | 0.024* | 0.991 | <0.001* | 1 | 0.217 | 1 | 0.22 |
| Thalamus | 57 | 21 | R | 13 (4.6) | −18 (4.3) | 9 (5.4) | 0.925 | 0.16 | 0.013* | 0.997 | 0.07 | 0.979 | 0.615 | 0.098 | 0.429 |

For each cluster we report: the number of foci falling within the cluster; the centroid coordinates in the MNI stereotaxic space; the standard deviation (sd) of the Euclidean distance from the centroid along the three axes; the p values associated with the binomial and Fisher's tests. Significant main and interaction effects are marked with an asterisk. E elderly, Y young.

detection was associated with cognitive-motor tasks including action observation tasks and motor imagery for body parts[54,55].

## Discussion

This study reviewed the effects of aging on motor neurophysiology, as assessed using functional imaging measures. We considered all fMRI/PET studies where both young and elderly subjects were tested with motor paradigms. We classified these experiments based on the specific nature of the task (paradigms involving overt motor execution or cognitive-motor tasks, such as motor imagination/observation paradigms) and on the level of performance achieved by the elderly participants (similar to young subjects or declined, in terms of accuracy or reaction times). We used this last factor as a "proxy" of task demands. Notably, we included experimental tasks involving different effectors, executed with the right and/or the left side and/or bilaterally, to test whether potential aging-related neurofunctional differences might occur regardless of the specific motor task and the specific effector/side.

Our results indicate that motor control is associated with significant aging-related changes in brain activity. These changes involve not only motoric brain regions but also posterior areas such as the occipito-temporal cortex. Importantly, some of these differences depend on the specific nature of the motor task and/or on the level of performance achieved by the participants. These results are discussed considering the current neurocognitive models of aging.

Before entering the details of our neurofunctional results, we want to give a brief qualitative overview of the behavioural patterns associated with aging in the motor domain, considering both accuracy and reaction times measures.

A decline in motor performance has been extensively described for older adults, in terms of movement coordination difficulties (for example, during bimanual movements, see ref. [56]), increased variability in action execution[56,57], slowing of movements: such problems affect both upper limbs movements as much as gait and balance[58–61]. In our paper, we focused on motor paradigms adopted for PET/fMRI experiments. We analysed the motor performance of elderly subjects compared with their younger counterparts, and we observed that behavioural decline in the elderly participants was reported in 21 out of 40 studies. This further testifies the degradation of motor functions associated with the physiological process of aging, whereby older adults showed increased reaction times and/or reduced accuracy during the execution of a variety of motor tasks.

Interestingly, the decline in motor performance is not directly related to the specific nature of the motor task. A chi-squared analysis failed to find a significant association between (the number of studies reporting) a declined performance and the specific experimental paradigms (broadly classified as explicitly implying a motor act or as "cognitive" in nature, such as motor imagery, action observation).

In the next paragraph, we will address these changes within the central nervous system, showing how older adults rely on more widespread central nervous system engagement for motor control than young adults.

We found specific age-related topographical differences in the neurofunctional patterns at the level of the left occipital lobes (calcarine fissure and superior occipital gyrus). These clusters were more frequently activated in the elderly group compared to their younger counterparts. We also reported reduced recruitment of the supramarginal gyrus and of sensorimotor brain regions in elderly group. Notably, this cluster was more frequently associated with the use of the contralateral part of the body for young subjects than for the older ones, independently from the

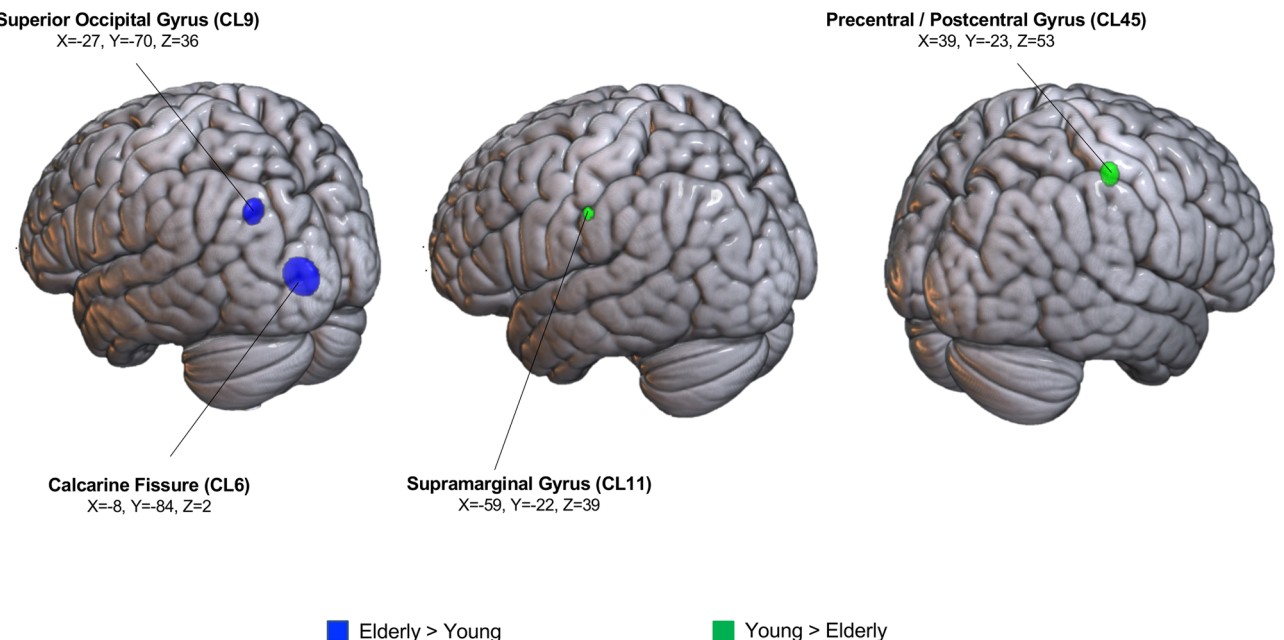

**Fig. 1 Clusters showing a significant main effect of group.** Clusters associated with elderly individuals are depicted in blue, whereas clusters specific for young subjects are depicted in green.

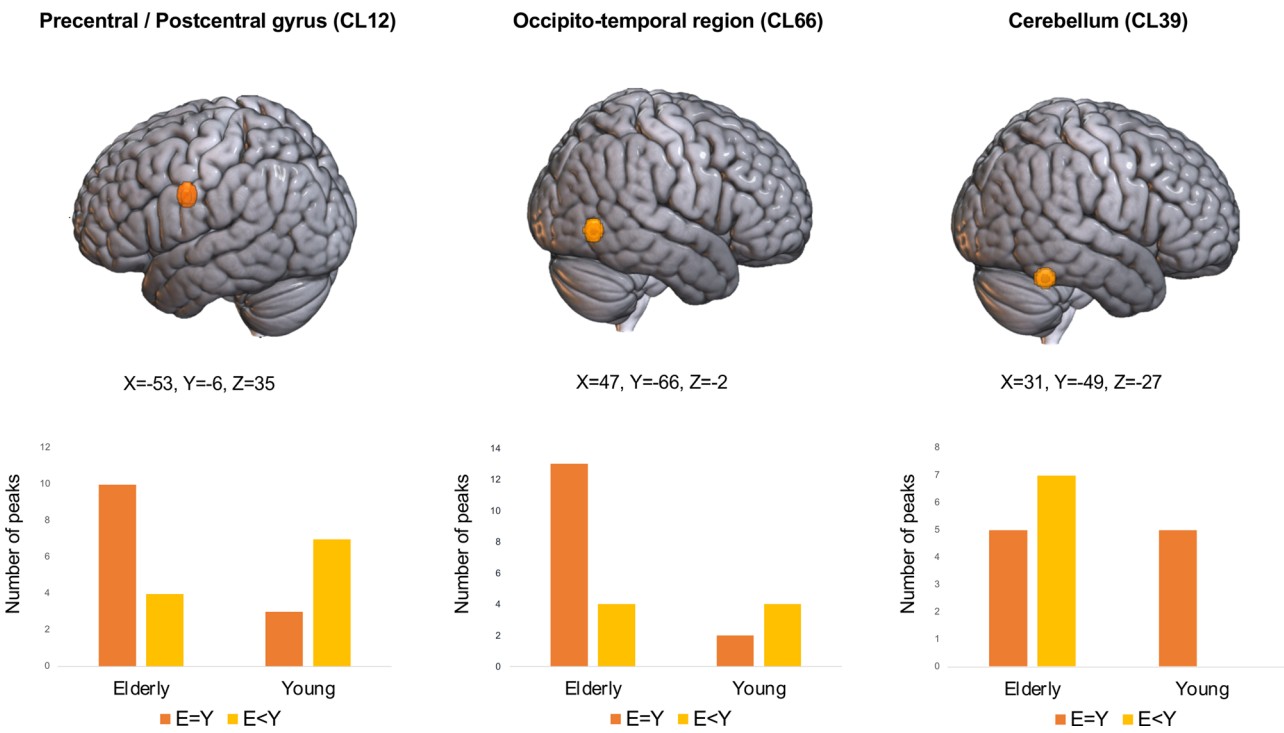

**Fig. 2 Clusters showing a significant group-by-performance interaction effect.** Top: Distribution of clusters showing a significant group-by-performance interaction effect. Bottom: Bar plot for the significant group-by-performance interaction, from left to right, in the left precentral/postcentral gyrus, in the right occipito-temporal cortex and in the right cerebellum. Orange bars indicate the number of peaks of each cluster associated with an equal performance between the elderly and the young group, yellow bars indicate the number of peaks of each cluster associated with a declined performance of the elderly group. Plots source data are provided in Supplementary Data 4.

level of performance achieved behaviourally. In other words, in older subjects the cluster contained a proportionally greater number of activation peaks associated with ipsilateral movements. This is in line with the so-called "age-related reduction of hemispheric lateralization" at the level of sensorimotor regions already highlighted by Turskey et al.[49] in a meta-analysis conducted on motor execution data.

Overall, these findings align with the idea that older adults demonstrate less selective recruitment of brain regions relative to young adults, possibly reflecting a breakdown in the functional segregation achieved after the complete central nervous system maturation[62,63]. This segregation might represent a process best characterized by an inverted U-shaped trajectory of maturation of visuomotor circuits, from typical development to maturation and

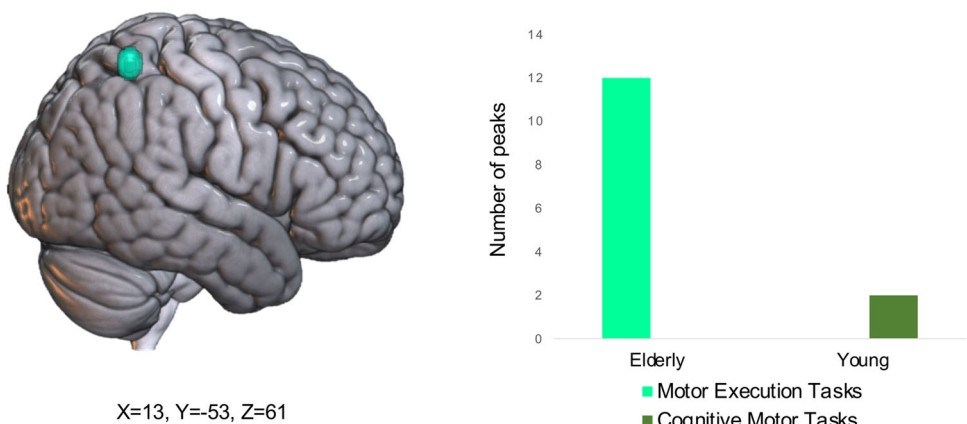

**Fig. 3 Cluster showing a significant group-by-performance interaction effect.** Light green bars indicate the number of peaks of the cluster associated with explicit motor execution tasks, dark green bars indicated the number of peaks associated with cognitive-motor tasks. Plots source data are provided in Supplementary Data 4.

then aging, whereby motor performance may initially rely also on visual inspection of what we do[39,64,65]. During adulthood, motor control becomes more independent from visual strategies and then back to the dependency on visual control at later stages of life.

Here, the additional occipital activations observed in elderly subjects suggest that they may use a mental visual imagery strategy even to perform explicit motor tasks. The results of the study of Coats and Wann[66] are in line with this hypothesis. These authors studied young and elderly subjects during reaching and grasping movements, using an apparatus that eventually obscured the target and the approaching hand after allowing an initial visual exploration. For the elderly subjects, both reaching and grasping were affected selectively when visualization of the hand was prevented, with additional reaching movements and longer adjustment times for the grasping phase of the movement[66]. Runnarong et al.[67] found similar findings, reporting that reach-to-grasp performance deteriorated with age, and the level of performance was specifically affected when the vision of the hand was occluded. Combined with our results, these findings suggest that elderly subjects are more reliant on visual feedback than the younger ones in tasks that require precise manual control.

The significant group-by-task interaction at the level of the precuneus supports this interpretation: elderly participants activated more frequently this specific region even during motor execution tasks, while their young counterparts recruit the same extra-striate area mainly during cognitive-motor tasks.

Taken together, our significant group effects indicating a more frequent recruitment of occipital regions in elderly subjects during the execution of a variety of motor tasks are in contrast with the hypotheses made by the HAROLD[30] and the PASA[33] pattern models, whereby aging should be accompanied by augmented and bilateral recruitment of the frontal lobes and reduced posterior activations. On the contrary, we found reduced brain activations only at the level of sensorimotor cortices.

It is important to emphasize that, beyond mere group effects, we also found significant interactions between factor age, the level of task performance achieved, and the specific nature of the task with respect to the presence of an overt motor output. These were localized in sensorimotor and occipito-temporal cortices and in the cerebellum. The functional meaning of these interactions

varied in the different regions. On the one hand, the interaction in the occipito-temporal cluster can be interpreted as a successful compensation. In this cluster, brain activity was higher in elderly participants when they achieved a level of performance similar to their younger counterparts, while the number of peaks associated with a deteriorated performance was significantly lower. On the other hand, the interaction in sensorimotor regions can be differently interpreted depending on the performance achieved by the older adults: the peaks forming this cluster derived from studies characterized by a declined behaviour, but also from experiments in which the elderly group reach the same performance level of younger subjects. Accordingly, there are indications that the activity in this region can reflect both successful and failed compensatory processes. This latter scenario is explained by the fact that this region is also significantly hypoactivated in the elderly subjects in the presence of a declined performance.

Finally, the interaction at the level of the cerebellum can be seen as an example of dedifferentiation: indeed, the higher number of peaks associated with the elderly subjects comes from studies where their performance was worse than the younger ones. This would be in line with what observed in previous studies: for example, Carp et al.[68] reported how motor distinctiveness, defined using multivariate pattern analysis, was reduced in older adults in a series of brain regions, including the cerebellum. However, there is one conceptual difficulty when considering the cerebellum in the context of a dedifferentiation hypothesis, given the motoric nature of the tasks behind this result. Indeed, the cerebellum contributes to motor performance in a dynamic manner, with a greater contribution during the learning phase of a new motor skill (see, for example ref. [69]). Accordingly, it is conceivable that the more frequent activation of the cerebellum in older adults (when they showed a declined performance) might be due to the fact that they perceived motor tasks as novel and less automatized, as when new motor learning is occurring or as a sign of a lost motor automaticity.

Overall, these results are in line with what hypothesized by the CRUNCH hypothesis[36]. Indeed, we showed that at relatively low levels of task demands/good performance, one can observe region-specific hyperactivations in older subjects (right occipito-temporal and left sensorimotor clusters). Interestingly, such compensatory overactivations depended on the specific nature of the

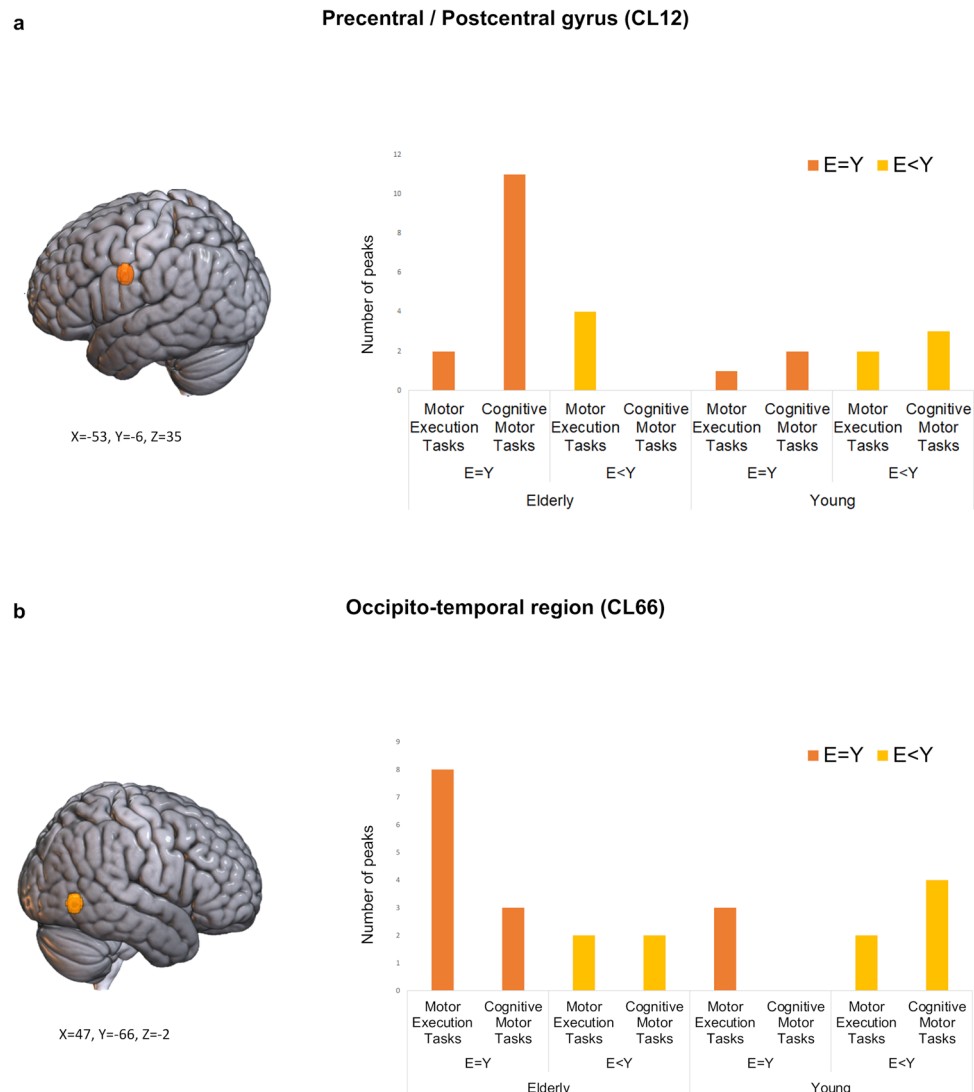

**Fig. 4 Clusters showing significant group-by-performance and task-by-performance interaction effects. a** Orange bars indicate the number of peaks of the Precentral/Postcentral gyrus cluster (CL12) associated with an equal performance between the elderly and the young group, while yellow bars indicate the number of peaks associated with a declined performance of the elderly group. **b** Orange bars indicate the number of peaks of the Occipito-Temporal cluster (CL66) associated with an equal performance between the elderly and the young group, while yellow bars indicate the number of peaks associated with a declined performance of the elderly group in. Plots source data are provided in Supplementary Data 4.

experimental task, with sensorimotor areas associated with motor execution tasks and posterior regions recruited for cognitive-motor tasks (see the exploration of the three-way interaction). With the increase of the task demands/decrease of performance, the attempt at compensation becomes less successful, and it can be defined as an unsuccessful attempt or an expression of dedifferentiation processes (right cerebellar cluster). Beyond a certain level of task demands, the elderly brain does not show sufficient activation levels, hence performance declines relative to the younger group (left sensorimotor cluster). Overall, this pattern also supports the idea that the CRUNCH and the dedifferentiation hypotheses might not be incompatible (see for example ref. [68]).

We wish to conclude with a word of caution: while the results of our meta-analysis broadly support the CRUNCH hypothesis[36], we are aware that to confirm our speculations further and fully support the model, this hypothesis should be formally tested in the same experimental context, for example by parametrically varying the level of task demands within the same experimental paradigm in the same sample of subjects (see ref. [70]).

Moreover, we should recognize that our approach cannot entirely test the HAROLD model and its adaptation to motor control (the so-called "sensorimotor lateralization hypothesis"), since we included experimental tasks executed with both body sides (and different effectors). This was done to investigate whether potential aging-related neurofunctional effects might occur regardless of the specific effector and/or the side used to execute the task. However, it should be acknowledged that most of the constituent studies employed hand movement tasks, which confines foci to a limited spot along the sensorimotor cortices.

Finally, when investigating individual differences among older adults, it is important to consider the limitations of cross-sectional designs compared with longitudinal designs. For example, older participants are typically recruited only from the subset of well-educated people aged in relatively good health and are free of brain disease, whereas the young adult samples with which they are compared are more heterogeneous. Cross-sectional study designs can also be contaminated by birth cohort effects, including inter-generational IQ increases.

Although they have their limitations, longitudinal study designs can avoid these problems.

## Methods

Our meta-analytical approach involves a series of analytical steps starting from the identification of the raw data (data collection and data preparation), followed by HC and statistical inferences on the clusters, which comprise a CCA. These procedures are described in detail below.

**Data collection and preparation**. We identified neuroimaging studies exploring the neural correlates of motor control during either motor execution tasks or non-execution cognitive-motor tasks (i.e., motor imagery, motor observation and motor prediction) in young and elderly individuals, using the following procedures.

First, we entered the following queries in PubMed (https://www-ncbi-nlm-nihgov.proxy.unimib.it/pubmed/): "fMRI and ageing and action", "fMRI and aging and action", "fMRI and older and action", "fMRI and age and action", "fMRI and ageing and [motor control]", "fMRI and aging and [motor control]", "fMRI and older and [motor control]", "fMRI and age and [motor control]", "fMRI and ageing and premotor", "fMRI and aging and premotor", "fMRI and older and premotor", "fMRI and age and premotor", "fMRI and ageing and motor", "fMRI and aging and motor", "fMRI and older and motor", "fMRI and age and motor", "PET and ageing and action", "PET and aging and action", "PET and older and action", "PET and age and action", "PET and ageing and [motor control]", "PET and aging and [motor control]", "PET and older and [motor control]", "PET and age and [motor control]", "PET and ageing and premotor", "PET and aging and premotor", "PET and older and premotor", "PET and age and premotor", "PET and ageing and motor", "PET and aging and motor", "PET and older and motor", "PET and age and motor", "neuroimaging and ageing and action", "neuroimaging and aging and action", "neuroimaging and older and action", "neuroimaging and age and action", "neuroimaging and ageing and [motor control]", "neuroimaging and aging and [motor control]", "neuroimaging and older and [motor control]", "neuroimaging and age and [motor control]", "neuroimaging and ageing and premotor", "neuroimaging and aging and premotor", "neuroimaging and older and premotor", "neuroimaging and age and premotor", "neuroimaging and ageing and motor", "neuroimaging and aging and motor", "neuroimaging and older and motor", "neuroimaging and age and motor". The initial set of studies included 20115 papers, updated to February 2021.

Second, after the removal of duplicates, we ran a preliminary selection based on the titles and abstracts of the papers, through which we included the studies that matched the following criteria: studies including both healthy young and older adults; studies reporting whole-brain activation peaks (no region-of-interest analyses) either from each group independently or from contrasts of the two groups; data reported using stereotactic coordinates (either MNI or Talairach atlases); task-based fMRI studies (no resting-state studies); univariate statistical analyses.

Conversely, we excluded the studies that matched the following criteria: studies that used neuroimaging methods other than task-based PET/fMRI studies, such as resting-state fMRI, PET, SPECT, or other non-fMRI procedures, to exclude variability across different neuroimaging findings; studies that assessed the effect of medication or other treatments without reporting fMRI data at baseline; studies analysed with a priori region of interest approach.

This selection, initially based on titles and then on abstracts, yielded to the identification of 161 candidate papers for the meta-analysis. Third, we made a further selection by inspecting the entire manuscripts and applying the aforementioned inclusion criteria in detail. Further, we conducted an up-to-date manual scan of the references of the selected articles, to ensure that all relevant papers had been included. For the suitable studies, we considered data derived from (i) within group simple effects and (ii) between group comparisons. We also incorporated within group data to have a more complete survey on whether a given brain region was differentially activated across groups, while still being active in each group above a given conventional threshold, or whether the region, besides being significantly associated with one group, it never reached statistically significant effects in the other group. The flowchart of the selection process is illustrated in Supplementary Fig. 4.

By applying such criteria, we included 40 papers[71–111], 142 contrasts and 1200 activation foci (857 peaks relative to elderly group and 343 peak relative to young group). A detailed description of the experimental paradigms is provided in Supplementary Data 1.

To arrange the dataset for the subsequent cluster composition analysis (CCA, see below for the details), each focus of activation was classified according to the three factors of interest: group (Elderly vs young), level of achieved behavioural performance (Elderly = young vs Elderly < young) and specific nature of the task (Motor execution paradigms vs cognitive-motor paradigms). All the Talairach coordinates were converted to MNI space using the Talairach to MNI transformation implemented in GingerALE[112,113], version 2.3.6. Twenty-eight out of 1270 foci were excluded from the dataset because they fell outside, even from the less conservative brain boundary mask of the GingerALE software.

The final dataset comprised 1349 participants, 616 elderly and 607 young participants. Please note that 126 participants cannot be assigned to the young/

elderly group since in the original studies participants were not divided in two groups and authors performed a regression analysis on a single group using the variable age as covariate (see refs. [103,104,111]). Even if some of the included papers were from the same group of authors[75,76,80–82,103,104,108–110], there was no overlapping of participants across the different studies.

The elderly group age range was 58-80, while the age range for the young group was 21-31.

**HC analysis and CCA**. To identify anatomically coherent regional effects, we first performed a HC analysis using the unique-solution clustering algorithm developed by Cattinelli et al.[114]. This method is implemented in a suite of MATLAB (2014a MathWorks) and C++ scripts called CluB (Clustering the Brain[115]). The CluB toolbox permits both to extract a set of spatially coherent clusters of activations from a database of stereotactic coordinates, and to explore each single cluster of activation for its composition according to the cognitive dimensions of interest. Crucially, this last step, called "cluster composition analysis", permits to explore neurocognitive effects by adopting a factorial-design logic and by testing the working hypotheses using either asymptotic tests, or exact tests either in a classic inference, or in a Bayesian-like context. This is something that cannot be done with GingerALE. Indeed, meta-analyses based on ALE are limited by the need of testing regional functional anatomical effects from highly homogeneous studies permitting, at the most, the evaluation of the neurofunctional differences (e.g., Group A > Group B) and commonalities (e.g., conjunction effect of Group A & Group B) between two classes of studies. The software cannot test more complex factorial models, the level of analysis needed for a complex neurocognitive scenario like the one behind aging and the nature of the task/level of the performance achieved.

Specifically, the CluB toolbox considers the squared Euclidean distance between each pair of foci included in the dataset. The clusters with minimal dissimilarity are recursively merged using Ward's criterion[116], to minimize the intra-cluster variability and maximizing the between-cluster sum of squares[114]. To impose a suitable a priori spatial resolution to our analyses, we set this to be 7 mm in terms of the maximum mean spatial variance within each cluster in the three directions (full width half maximum, FWHM). The centroid coordinates of each resulting cluster were then labelled according to the automatic anatomic labelling (AAL) and then controlled by visual inspection on the MRIcron[117] visualization software.

The output of the HC analysis was then entered as an input for the subsequent CCA. This procedure allows a post-hoc statistical exploration of each cluster by computing, within each cluster, the proportion of foci belonging to different levels of a variable of interest. Such proportion is then compared with a target proportion, which, in our case, is extracted from the overall distribution of foci classified according to our factors of interest in the whole dataset (prior likelihood, PL). First, we ran a CCA to explore the main effects of group, nature of the task, and level of performance. This composition analysis was done by running a two-sided binomial test on the proportion of foci associated with each level of the three factors within each cluster. For example, if a cluster X had a cardinality of $N = 20$ and included 15 foci associated with the level "Elderly" of the "group" factor, CluB computes the proportion 15/20 (i.e., 0.75) and compares it with the theoretical proportion computed over the entire dataset (e.g., young/elderly = 377/650 = .58). Hence, (a) the Prior Likelihood represents the probability of success under the null hypothesis and (b) a significant binomial test ($p < .05$) indicates that the proportion of activation peaks included in that specific part of the brain is higher than the proportion computed all over the brain. Afterwards, to test for interaction effects (group-by-task, group-by-performance, and task-by-performance), we performed a series of two-sided Fisher's exact tests[118] on the empirical peak-distribution within each cluster. Despite not being associated with a formal statistical test, the same procedure was applied to explore, descriptively, eventual three-way interaction effects (i.e., group-by-task-by-performance interaction).

**Validation of the spatial relevance of each cluster using the ALE procedure**. As the HC procedure does not provide a statistical test of the spatial significance of the resulting clusters, this can be compensated for by searching for spatial convergence between the clustering solution and the results of an activation likelihood estimate ALE-based meta-analysis on the same overall dataset (see, for example refs. [118,119]). For the spatial cross-validation with ALE we employed the Turkeltaub Non-Additive method[113], with the general statistical threshold set to $p < 0.05$ FWE corrected (cluster-level). The resulting maps were overlapped with the HC map with the "intersection" function in the software MRIcron (https:// www.nitrc.org/projects/mricron). Only the clusters that fell in this intersection map were then considered for further analyses (the CCA) and discussion.

**Statistics and reproducibility**. The details about statistics used in different data analyses performed in this study are given in the methods section.

In short, we first performed a hierarchical clustering analysis using the unique-solution clustering algorithm developed by Cattinelli et al.[114], to identify anatomically coherent regional effects. This method is implemented in a suite of MATLAB (2014a MathWorks) and C++ scripts called CluB (Clustering the Brain[115]).

The output of the HC analysis was then entered as an input for the subsequent CCA. This procedure allows a post-hoc statistical exploration of each cluster by

computing, within each cluster, the proportion of foci belonging to different levels of a variable of interest performing a series of two-sided binomial tests.

Furthermore, we performed a series of two-sided Fisher's exact tests[118] on the empirical peak-distribution within each cluster in order to test for interaction effects (group-by-task, group-by-performance, and task-by-performance).

Despite not being associated with a formal statistical test, the same procedure was applied to explore, descriptively, eventual three-way interaction effects (i.e., group-by-task-by-performance interaction).

As the HC procedure does not provide a statistical test of the spatial significance of the resulting clusters, this can be compensated for by searching for spatial convergence between the clustering solution and the results of an ALE-based meta-analysis on the same overall dataset. We employed the Turkeltaub Non-Additive method[113], with the general statistical threshold set to $p < 0.05$ FWE corrected (cluster-level).

**Reporting summary**. Further information on research design is available in the Nature Research Reporting Summary linked to this article.

## Data availability
Source data are provided in Supplementary Data 1–4. Any additional datasets generated during and/or analysed during the current study are available from the corresponding author on reasonable request.

## Code availability
Data analyses were conducted using GingerALE (https://brainmap.org/ale/) and CluB (https://osf.io/4b2pc/wiki/home/) software.

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

## Acknowledgements

This paper was supported by a grant funded by the Italian Ministry of Health to E.P. (Ricerca Corrente; Project L3025). We are grateful to the authors of the original empirical work that was submitted to meta-analysis here. Without their work, we would have not been able to produce this manuscript.

## Author contributions

L.Z., M.M. and E.P. reviewed the data for the meta-analyses, performed the analyses and drafted the manuscript.

## Competing interests

The authors declare no competing interests.
