## [Transparent Peer Review File · Communications Biology]

Reviewers' comments:

Reviewer #1 (Remarks to the Author):

This study reports on age-related changes to the functional neuroanatomy subserving motor and motor imagery tasks using a combination of quantitative meta-analysis approaches. I particularly appreciated the authors' attempts to determine whether these age-related changes in (concordance of) brain activity are moderated by whether performance differed by age or the type of task being performed. I would not consider these critical questions in the field, but they do merit consideration. However, there were many theoretical and methodological difficulties I had with this manuscript that deter me from recommending it for publication. Namely:

There is a substantial gap in the literature review. The authors seem to rely heavily on literature of cognitive aging, even though there already exists a wide literature on motor aging, including a review and meta-analysis already:

Please see work by Rachael Seidler, including a review:

Bernard JA, Seidler RD. 2012. Evidence for motor cortex dedifferentiation in older adults. *Neurobiol Aging*. 33:1890–1899.

And a quantitative ALE meta-analysis may be found here:

Turesky TK, Turkeltaub PE, Eden GF. 2016. An Activation Likelihood Estimation Meta-Analysis Study of Simple Motor Movements in Older and Young Adults. *Front. Aging Neurosci*. 8:238.

The second citation repudiates statements in the manuscript such as this one: "this is the first meta-analytical attempt to summarize the previous findings on aging in the domain of motor control."

The existence of these studies also attenuates the novelty and significance of this work.

There is also some terminology that is prevalent in the motor literature that I did not find here. Namely, de-differentiation, which the authors describe but don't name explicitly. In addition to the above review, please see here for motor studies that examine age-related changes and have findings that are interpreted in this context:

Carp J, Park J, Hebrank A, Park DC, Polk TA. 2011. Age-related neural dedifferentiation in the motor system. *PLoS One*. 6:e29411.

Loibl M, Beutling W, Kaza E, Lotze M. 2011. Non-effective increase of fMRI-activation for motor performance in elder individuals. *Behav Brain Res*. 223:280–286.

Riecker A, Gröschel K, Ackermann H, Steinbrink C, Witte O, Kastrup A. 2006. Functional significance of age-related differences in motor activation patterns. *Neuroimage*. 32:1345–1354.

I think the authors used the last one in their analysis, but there are many additional studies of the aging motor system that the authors did not use or reference. I understand completely that many of these would not be usable based on inclusion/exclusion criteria. However, describing their findings would be helpful.

It may also be beneficial to report on the motor system as characterized by functional neuroimaging in general. For instance, there is a meta-analysis of finger-tapping that reports on this:

Witt ST, Laird AR, Meyerand ME. 2008. Functional neuroimaging correlates of finger-tapping task variations: an ALE meta-analysis. *Neuroimage*. 42:343–356.

One theory the authors describe in the introduction in the context of cognitive function relates to bilaterality as a function of age. Many studies of the aging motor system also have found greater recruitment of cortical motor areas ipsilateral to the side of movement in elderly adults (and/or greater bilaterality). It's unclear why this is not discussed.

The authors offer "educated guesses" in place of hypotheses, which does not sound particularly scientific and I'm having a little trouble understanding why one would predict over activation of prefrontal cortices for motor tasks. In other words, why would age-related differences on motor tasks manifest similarly to age-related differences on cognitive tasks, especially with so many empirical studies (such as those referenced above) reporting age-related differences in motor areas.

If there are 40 studies total, with 9 studies having greater reaction times for elderly and 14 studies having reduced accuracy for the elderly, then why do only 21 studies report behavioral declines?

The chi-square test is not clear to me in a couple respects. First, the factors indicate performance, but seem not to specify whether accuracy or response time or both were used here. Second, they indicate that the factors are performance and task type, but then later on write of the result "age and nature of the task."

It would be helpful to have much more of the info from Table S1, particularly about the Task and Experimental paradigm, in the main text, prior to the Results. In this vein, some methods ahead of the results would be helpful for interpreting the results. For instance, ALE is written first as an abbreviation (presumably because the methods section, where it is written out, generally precedes the results section).

The diversity of tasks may be problematic for identifying concordance in motor areas, especially for primary sensorimotor cortex. This area is quite large and mostly considered to be organized according to body parts (somatotopy). Thus, combining studies that examine finger movements with studies that examine speech may mean using foci that are distantly spaced and unlikely to converge in statistical analyses. This has direct implications for testing the age-related lateralization theory.

In this vein, with motor tasks, laterality is a central issue as typically in young adults movements made with one side of the body elicit robust activation in contralateral cortical motor areas and in ipsilateral anterior cerebellum. It would be helpful to specify prior to the results whether the tasks are for right hands, left hands, unimanual or bimanual, dominant, non-dominant hand, etc. and whether the participants in these tasks are right- or left-handed. The results were difficult for me to contextualize without this information.

At line 399, the authors label a finding as premotor, but the finding appears to be in pre/postcentral gyrus, which is generally more considered primary sensorimotor or primary somatomotor cortex. How was the functional label of premotor determined? Care should be taken with these labels. There are studies that help with appropriate functional labeling:

Mayka M A, Corcos DM, Leurgans SE, Vaillancourt DE. 2006. Three-dimensional locations and boundaries of motor and premotor cortices as defined by functional brain imaging: a meta-analysis. *Neuroimage*. 31:1453–1474.

At line 421, the characterization of "with increasing cognitive load" is somewhat perplexing. My understanding was that the finding alluded to related to performance, not task complexity.

What standards for thresholds were used in identifying foci to include in the meta-analysis. For instance, some studies (maybe more common in older studies) will report on clusters that don't survive correction for multiple comparison.

It's not clear whether the coordinates from within-group data were treated differently from coordinates from between-group data. Using foci from a cluster that is present in one group but not the other is not the same as using foci from between-group clusters, which the authors seem to acknowledge.

The same groups seem to have multiple studies from around the same time included in the meta-analysis (e.g., Van Impe and Heuninckx). Just to be clear, these do not reflect overlapping groups of participants?

More minor:

The first part of title is a little strange "motor brain"??

The abstract alternates between results and interpretations

Was the ALE map threshold voxel- or cluster-level corrected?

At line 542, what is "district"?

There doesn't seem to be a genuine conclusion. the manuscript ends with limitations.

There are many grammatical and spelling mistakes, and the language at times is difficult to follow; e.g., "none of these models has not taken into great consideration the motor control dimension" or "At variance with what can be done in a novel empirical experiment, in which the variables under examination are controlled by the experimenter, meta-analyses are more observational in nature."

Reviewer #2 (Remarks to the Author):

In the present paper, Zapparoli et al. performed a quantitative meta-analysis of 40 brain imaging studies to assess the effects of aging on motor control. Results identified higher activation in the occipital cortex and lower activation the sensorimotor cortex for older compared to younger participants, as well as age by performance interactions in sensorimotor, occipito-temporal, and cerebellar regions. Findings were interpreted in line with major theories of neurocognitive aging and found to support mainly the Compensation Related Utilization of Neural Circuits Hypothesis (CRUNCH). This meta-analysis is a needed contribution to an understudied line of investigation. However, a few aspects need to be addressed before the contribution of this paper can be fully assessed, mainly regarding the clarity of the methods and presentation of the results, and a discussion section that seems somewhat unfocused and overly speculative.

1. It is not entirely clear what the advantage of the current method is, compared to the widely-used ALE procedure, and why the authors preferred this approach if they ended up "masking" with ALE results? Please clarify.

2. In the methods section, for the 3-way interaction, it is stated that it was not "associated with a formal statistical test". Yet, in the results, the authors talk about a group x task x performance interaction.

3. It is unclear what the "undifferentiated clusters" are. For instance, CL39 (mentioned in the text) does not show any significant effects in Table 2. Please clarify. Also, please check the correspondence between main text and Table 2.

4. In its present form, the discussion seems to be overly long and lacking focus, especially the "Neurophysiological aging effects" part. I think more emphasis should be given to how the present findings specifically provide support for or against the theories of neurocognitive aging mentioned in the introduction.

Minor comments:

- I found Table 2 difficult to read, mainly due to its width. I would suggest listing the coordinates for both hemispheres in the same column, preceded by a column with L/R entries.
- The text needs careful proofreading as there are a number of typos and awkward phrases that affect comprehension.

Reviewer #1

This study reports on age-related changes to the functional neuroanatomy subserving motor and motor imagery tasks using a combination of quantitative meta-analysis approaches. I particularly appreciated the authors' attempts to determine whether these age-related changes in (concordance of) brain activity are moderated by whether performance differed by age or the type of task being performed. I would not consider these critical questions in the field, but they do merit consideration.

A.R. We would like to thank the reviewer for her/his overall positive appreciation of our efforts.

However, there were many theoretical and methodological difficulties I had with this manuscript that deter me from recommending it for publication. Namely:

There is a substantial gap in the literature review. The authors seem to rely heavily on literature of cognitive aging, even though there already exists a wide literature on motor aging, including a review and meta-analysis already:

- **Bernard JA, Seidler RD. 2012. Evidence for motor cortex dedifferentiation in older adults. *Neurobiol Aging*. 33:1890–1899.**
- **Turesky TK, Turkeltaub PE, Eden GF. 2016. An Activation Likelihood Estimation Meta-Analysis Study of Simple Motor Movements in Older and Young Adults. *Front. Aging Neurosci*. 8:238.**

The second citation repudiates statements in the manuscript such as this one: "this is the first meta-analytical attempt to summarize the previous findings on aging in the domain of motor control." The existence of these studies also attenuates the novelty and significance of this work.

A.R. We agree with the Referee that our statement needs a better specification. Indeed, may we have considered motor function (elementary and cognitive) as a whole, our statement would not be justified. However, our claim of originality still holds given the following specifications: for the first time our meta-analysis considers not only the aging factor, but also the nature of the task (cognitive motor tasks or motor execution tasks), the level of performance (equal or not equal to the younger counterparts, performance being used as a proxy of task demands) and their interactions and how these impact on brain activity. Crucially, our meta-analytical approach allowed us to explore these effects by adopting a factorial design - not affordable using the ALE technique - and by testing the working hypotheses using either asymptotic tests or exact tests. As far as the literature cited by the Referee, we remark that the meta-analysis he refers to considered only motor execution studies with the same paradigm (regularly paced right-hand finger movements). We value this previous evidence, and we now cite it, together with the TMS study of Bernard's et al (2012). This last paper was mentioned in the last section to discuss our findings in the context of the *dedifferentiation hypothesis* (see the next comment for more details). To conclude on this issue, the reviewer will hopefully agree that, bearing in mind the specifications mentioned above, our new study retains a sufficient level of novelty, either in terms of technique or in terms of scope.

On page 3, we now write:

In the present paper, we address this specific topic by reviewing with a quantitative meta-analysis the available evidence derived from task-based brain imaging activation on how the physiological process of aging affects motor control. Crucially, we considered not only explicit motor execution tasks, but also cognitive motor tasks (e.g., motor imagery, motor prediction paradigms, typically used to investigate motor functions, see for example Gandola et al. 2017 or Zapparoli et al. 2016) and the achieved level of behavioral performance: this

last aspect was taken as an indirect index of the task demands. To the best of our knowledge, this is the first meta-analytical attempt to summarize the previous findings on aging in the domain of motor control with this specific approach.

Moreover, on page 5, we wrote:

Only one meta-analysis explored the effects of aging on the neurofunctional correlates of motor control: Turesky and colleagues (2016) conducted an activation likelihood estimation (ALE) meta-analysis on older adults and young adults all performing paced right-hand finger movement tasks. The authors found that elderly subjects activated more frequently the right sensorimotor cortex, the right supramarginal gyrus, the medial premotor cortex, and the right posterior cerebellum. They concluded that older adults more frequently activate ipsilateral brain regions for right-hand finger movements (Turesky et al., 2016).

However, this paper did not explore the possible interactions of the factor aging with the level of performance achieved by the subjects and the nature of the motor task, something that we aimed at with the present study, where we considered also motor cognitive tasks (e.g., motor imagery, action prediction, action observation...). Moreover, the mentioned meta-analysis only focused on one specific (explicit) motor task, performed with the same limb; here, we aimed at investigating whether such findings might be generalized independently from the specific motor task and the body district involved in such tasks.

Finally, on page 12, we wrote:

On the other hand, the dedifferentiation hypothesis posits that age-related brain functional changes might indicate a generalized diffusion of brain activity attributable to deficits in neurotransmission, which in turn causes a loss of neural specialization. Based on this hypothesis, hyperactivations should be larger in poor than in successful motor performers (Li and Lindenberger, 1999). The dedifferentiation hypothesis has been previously supported by different experiments. For example, Loibl et al. (2011) observed age-related increased activations in ipsilateral motor areas (M1 and SMA) during various movements, that were negatively correlated with motor performance. Similar results were found by Bernard et al. (2012) with transcranial magnetic stimulation, where the authors reported more diffuse motor cortical activations in the hemisphere contralateral and ipsilateral to the recorded motor evoked potentials. Crucially, such broader activations correlated with higher reaction times (Bernard et al., 2012).

There is also some terminology that is prevalent in the motor literature that I did not find here. Namely, de-differentiation, which the authors describe but don't name explicitly. In addition to the above review, please see here for motor studies that examine age-related changes and have findings that are interpreted in this context:

- **Carp J, Park J, Hebrank A, Park DC, Polk TA. 2011. Age-related neural dedifferentiation in the motor system. PLoS One. 6:e29411.**
- **Loibl M, Beutling W, Kaza E, Lotze M. 2011. Non-effective increase of fMRI-activation for motor performance in elder individuals. Behav Brain Res. 223:280–286.**
- **Riecker A, Gröschel K, Ackermann H, Steinbrink C, Witte O, Kastrup A. 2006. Functional significance of age-related differences in motor activation patterns. Neuroimage. 32:1345–1354.**

I think the authors used the last one in their analysis, but there are many additional studies of the aging motor system that the authors did not use or reference. I understand completely that many of these would not be usable based on inclusion/exclusion criteria. However, describing their findings would be helpful.

A.R. As the referee correctly pointed out, the first two mentioned studies did not meet our inclusion criteria (the first one used an MVPA approach, while the second one did not report all the stereotactic coordinates of the effects of interest, and it was based on ROIs-oriented analyses). The details of our literature research are described in the methods sections and in Figure S4.

However, the reviewer is correct that we should pay tribute to anyone who already suggested data or interpretations of great relevance and perhaps consistent with our own. We normally do this mostly in the discussion section of our meta-analyses, as the single empirical studies represent the “raw data” for a meta-analysis. To this end and, we carefully scrutinized the literature, and we now emended this limitation.

On pages 12-13, we wrote:

Modifications of fMRI patterns in the elderly, **in terms of overactivations**, can be interpreted as evidence of two major hypotheses considering the behavioural performance of the elderly participants.

On the one hand, the **compensation hypothesis** predicts that age-related increases in brain activation, as well as the recruitment of additional areas, compensate for various neural/behavioral deficits^{15,27,60-62}. According to the compensation hypothesis, the **overactivations would be larger in good than in poor performers**. Compensatory patterns have been documented for several cognitive domains (e.g., working memory, episodic memory retrieval, perception, inhibitory control, etc.). This phenomenon was initially observed in the prefrontal cortex; moreover, compensatory processes have been described as reduced inter-hemispheric asymmetries for tasks associated with strongly lateralized fMRI patterns in younger participants²⁷. This overall pattern is well captured by the *HAROLD model*⁶⁰.

A similar compensatory hypothesis is covered by the *PASA model*²⁸, whereby aging is associated with a significant increment of the frontal lobes' activation and reduced neural activity in posterior areas (mainly occipito-temporal cortices). Age-related reductions in occipital activity have been attributed to deficits in sensory processing, while age-related increased prefrontal activity would represent an attempt to compensate for these deficits.

On the other hand, the **dedifferentiation hypothesis** posits that age-related brain functional changes might indicate a generalized diffusion of brain activity attributable to deficits in neurotransmission, which in turn causes a loss of neural specialization. This hypothesis suggests hyperactivations should be larger in poor than in successful motor performers⁶³. Several experiments have previously supported the dedifferentiation hypothesis. For example, Loibl et al.⁶⁴ observed age-related increased activations in ipsilateral motor areas (MI and SMA), negatively correlated with motor performance⁶⁴. Similar results were found by Bernard et al.⁶⁵ with transcranial magnetic stimulation, reporting for the elderly subjects more diffuse motor cortical activations in the hemisphere contralateral and ipsilateral to the recorded motor evoked potentials. Crucially, such broader activations correlated with higher reaction times⁶⁵.

However, these hypotheses seem insufficient to capture the dynamics of compensation in elderly subjects when considering the specific task demands (inferred by the level of performance achieved). These may change in an age-dependent manner as summarized by the CRUNCH hypothesis and the posited group by performance interactions.

This is what we report here for the first time with our meta-analysis in the domain of motor control. We found significant interactions in sensorimotor, cerebellar, and occipito-temporal cortices when considering, together with the factor age, the level of task performance achieved, and the specific nature of the task. The functional meaning of these interactions varied in the different regions. On the one hand, the interaction in the **occipito-temporal cluster** can be interpreted as a **successful compensation**. In this cluster, brain activity was higher when elderly participants achieved a level of performance similar to their younger counterparts, while the number of peaks associated with a deteriorated performance was significantly lower.

On the other hand, the interaction in sensorimotor regions can be differently interpreted depending on the performance achieved by the older adults: the peaks forming this cluster derived from studies characterized by a declined behavior, but also from experiments in which the elderly group reach the same performance level

of younger subjects. In this sense, there are indications that the activity in this region can reflect both successful and failed compensatory processes. This latter scenario is explained by the fact that this region is also significantly hypo-activated in the elderly subjects in the presence of a declined performance.

Finally, the interaction at the level of the cerebellum could be seen as an example of dedifferentiation: indeed, the higher number of peaks associated with the elderly subjects comes from studies where their performance was worse than the younger ones. This would be in line with what observed in previous studies: for example, Carp and colleagues (2011) reported how motor distinctiveness was reduced among older adults in a series of brain regions, including the cerebellum⁶⁶. However, there is one conceptual difficulty when considering the cerebellum in the context of a dedifferentiation hypothesis, given the motoric nature of the tasks behind this result.

Indeed, the cerebellum contributes to motor performance in a dynamic manner, with a greater contribution during the learning phase of a new motor skill (see, for example⁶⁷). Accordingly, it is conceivable that the more frequent activation of the cerebellum in older adults (when they showed a declined performance) might be due to the fact that they perceived motor tasks as novel and less automatized, as when new motor learning is occurring.

To summarize, these results indicate that at relatively low levels of task demand/good performance, one can observe region-specific hyperactivations in older subjects (right occipito-temporal and left sensorimotor clusters). Interestingly, such compensatory overactivations depended on the specific nature of the experimental task, with sensorimotor areas associated with motor execution tasks and posterior regions recruited for cognitive motor tasks (see the exploration of the three-way interaction). With the increase of the task demand/decrease of performance, the attempt at compensation becomes less successful, and it can be defined as an unsuccessful attempt or an expression of dedifferentiation processes (right cerebellar cluster). Beyond a certain level of task demand, the elderly brain does not show sufficient activation levels, hence performance declines relative to the younger group (left sensorimotor cluster). This is in line with what hypothesized by the CRUNCH hypothesis.

Overall, this pattern supports the idea that the CRUNCH and the dedifferentiation hypotheses might not be incompatible (see for example⁶⁶). Indeed, the dedifferentiation hypothesis predicts a reduction in performance together with an increase in activation, thus resembling the middle part of the relationship between task demands and fMRI activation hypothesized by the original CRUNCH hypothesis.

It may also be beneficial to report on the motor system as characterized by functional neuroimaging in general. For instance, there is a meta-analysis of finger-tapping that reports on this:

- **Witt ST, Laird AR, Meyerand ME. 2008. Functional neuroimaging correlates of finger-tapping task variations: an ALE meta-analysis. *Neuroimage*. 42:343–356.**

A.R. We discuss about this paper in the introduction section of the revised paper.

On pages 5-6, we wrote:

We combined a hierarchical clustering algorithm with the coordinate-based activation likelihood estimation (ALE). The former allowed us to assess the data according to a factorial design, while the ALE method allowed us to assess the spatial significance of the clusters identified with hierarchical clustering. We assessed whether (i) motor control is associated with aging-related meaningful changes in brain activity, (ii) such modifications depend on the level of task demand (inferred by the level of behavioural performance achieved by the elderly subjects) and/or on the specific nature of the motor task, (iii) how these changes can be interpreted in the light of the neurocognitive models of ageing.

In which brain regions should we expect such effects? One possibility is the involvement of cortical and subcortical areas typically associated with motor control, such as the primary sensorimotor cortices, the

supplementary motor area, premotor cortex, basal ganglia, and cerebellum⁴². Other possible brain candidates are the regions involved in cognitive motor tasks, such as fronto-parietal networks activated during motor imagery tasks⁴³, but also occipito-temporal regions typically recruited during action prediction and/or action observation paradigms⁴⁴.

One theory the authors describe in the introduction in the context of cognitive function relates to bilaterality as a function of age. Many studies of the aging motor system also have found greater recruitment of cortical motor areas ipsilateral to the side of movement in elderly adults (and/or greater bilaterality). It's unclear why this is not discussed.

A.R. The original theory proposed by Cabeza and colleagues (2002) considered the prefrontal cortex as the key neural correlate of aging-related neurofunctional changes. We thought this might apply also to motor functions, since the prefrontal cortices are involved in motor learning (see for example Jueptner et al., 1997, *Anatomy of motor learning. I. Frontal cortex and attention to action*). However, we agree with the referee that one possible extension of such theory to the domain of motor control would contemplate a more bilateral sensorimotor recruitment due to greater ipsilateral activation of areas M1/S1. We discuss this specific topic in the revised version of the manuscript.

We modified the text accordingly and on page 6 we wrote:

On the one hand, the HAROLD model would be confirmed by an overactivation of the prefrontal cortices in elderly participants, specifically in association with a good behavioural performance. Since we consider here motor control functions, one possible extension of this model would imply an increased bilateral recruitment of sensorimotor regions (cfr Turesky et al. ⁴¹).

The authors offer "educated guesses" in place of hypotheses, which does not sound particularly scientific and I'm having a little trouble understanding why one would predict over activation of prefrontal cortices for motor tasks. In other words, why would age-related differences on motor tasks manifest similarly to age-related differences on cognitive tasks, especially with so many empirical studies (such as those referenced above) reporting age-related differences in motor areas.

A.R. Please see our previous comment on the same issue. On the basis of the referee's comment, we changed our wording on page 6:

Admittedly we did not commit ourselves to any of the aforementioned models from the outset: indeed, we consider these not to be necessarily mutually exclusive. Yet, we had clear the specific findings that would be in support of any given theory.

If there are 40 studies total, with 9 studies having greater reaction times for elderly and 14 studies having reduced accuracy for the elderly, then why do only 21 studies report behavioral declines?

A.R. Thanks for this comment. This is because there were different studies reporting both RTs and accuracy.

We now specify this on page 7:

Twenty-one out of 40 studies described a behavioral decline in the elderly participants, suggesting a general deterioration of motor functions associated with the physiological process of aging: older adults showed increased reaction times (RTs, 5 studies), or reduced accuracy (12 studies), or both increased RTs and reduced accuracy (4 studies) during the execution of different motor tasks (see Table 1 and Table S1).

The chi-square test is not clear to me in a couple respects. First, the factors indicate performance, but seem not to specify whether accuracy or response time or both were used here. Second, they indicate

that the factors are performance and task type, but then later on write of the result "age and nature of the task."

A.R. We considered both accuracy and response times for the chi-squared analysis. We are sorry, there was a typo: in this analysis we considered the level of performance achieved by the elderly subjects (equal or declined with respect to younger participants) and the nature of the task, to investigate whether there was a more frequent behavioural decline in one specific task (motor execution/motor cognitive tasks).

We corrected the text and on page 7 we wrote:

To assess whether impaired performance in older adults was more frequently associated with the specific nature of the task, we performed a chi-square analysis (Factor1, performance: Elderly=Young/Elderly<Young, considering both accuracy and RTs data; Factor2, nature of the task: motor execution tasks/cognitive motor tasks). The chi-square test of independence showed that there was no significant association between the level of behavioral performance achieved by the elderly participants and nature of the task, $X^2(1, N=40) = 1.58, p = 0.21$.

It would be helpful to have much more of the info from Table S1, particularly about the Task and Experimental paradigm, in the main text, prior to the Results. In this vein, some methods ahead of the results would be helpful for interpreting the results. For instance, ALE is written first as an abbreviation (presumably because the methods section, where it is written out, generally precedes the results section).

A.R. We included more information about the Experimental Paradigms in Table S1, as well as in the main text prior to the results (in the introduction section). Please note that we did not go deeper in the introduction since this is a systematic literature review and the studies included were based on the keywords/inclusion criteria listed in the methods section. In other words, the list of the studies included in the analysis is part of the results. Communications biology requires that the methods should come after the discussion, so it might be strange to add information about the paradigms before having spelled out the criteria for including the studies in the meta-analysis.

On page 5 we now specify:

In this study our aim was to investigate the effects of aging on the neurofunctional correlates of motor control, testing the different neurocognitive theories of aging with a meta-analytical approach. Besides the aging process, we considered two other factors: the specific nature of the tasks (motor execution tasks or cognitive motor tasks) and the tasks demand: this was inferred by the level of performance achieved by the elderly subjects, similar to young participants or inferior. We focused on evidence coming from fMRI/PET activation studies on groups of young and elderly individuals. As the reader will see below, we identified neuroimaging studies either on **explicit motor execution tasks** (i.e., simple motor paradigms such as finger-tapping/opposition tasks, bimanual or face/mouth movements; more complex paradigms such as repeated flexion-extension movements, force modulation tasks, drawing tasks) or **cognitive motor tasks** (i.e., motor imagery, motor observation and motor prediction). These tasks were executed mainly by right-handed participants.

The diversity of tasks may be problematic for identifying concordance in motor areas, especially for primary sensorimotor cortex. This area is quite large and mostly considered to be organized according to body parts (somatotopy). Thus, combining studies that examine finger movements with studies that

examine speech may mean using foci that are distantly spaced and unlikely to converge in statistical analyses. This has direct implications for testing the age-related bilateralization theory.

A.R. The reviewer is correct but her/his concern on the possibility of finding meaningful results within areas M1/S1 is justified only in part: it is true that the considered studies involved a variety of body segments. We now acknowledge this as a potential limitation as far as the possibility of mapping the aging effects within somatotopically organized areas like area M1/S1 if analysed as a single region using a large ROI. However, the hierarchical clustering technique used was based on a spatial kernel of 7 mm FWHM applied on 40 studies. To us, the spatial resolution combined with the sample size were sufficient for the identification of separate clusters, if any, along the M1/S1 cortical strip with a suitable chance of identifying clusters falling within a specific somatotopic territory. Once the clusters were identified, it became possible to explore the contributing peaks as far as the body segments that led to their identification. This is what happened and what we did: in a new Table (Table S3), we now specify the body segments involved in the identification of the clusters. As it happened, only two clusters falling M1/S1 were associated with aging or relevant interactions (Clusters 12 and 45). In none of these clusters there was an obvious association with a specific body segment. This is not as surprising because the different body districts show a good deal of overlap at the cortical boundaries between body segments within pre- and post-central cortices. In other two clusters, more obviously falling within area M1 for the hand, there were no group effects, nor interaction (Clusters 27 and Clusters 29).

This is now specified on page 11:

Notably, we included experimental tasks involving different effectors, executed with the right and/or the left side and/or bilaterally, to test whether potential aging-related neurofunctional differences might occur regardless of the specific task and the specific effector/side. However, the spatial resolution of our analyses permitted also to assess whether a specific effect could have arisen in a subdivision of M1/S1 mapping the body segment mostly involved in the tasks that generated the activation peaks.

In this vein, with motor tasks, laterality is a central issue as typically in young adults movements made with one side of the body elicit robust activation in contralateral cortical motor areas and in ipsilateral anterior cerebellum. It would be helpful to specify prior to the results whether the tasks are for right hands, left hands, unimanual or bimanual, dominant, non-dominant hand, etc. and whether the participants in these tasks are right- or left-handed. The results were difficult for me to contextualize without this information.

A.R. We added this information in the introduction section and in Table 1. Moreover, we considered this specific issue as a potential limitation in the discussion section, with respect to the so-called “bilateralization theory” (see the previous comment).

However, for each cluster in which we observed a significant “Group” main or interaction effect, we assessed whether there was a group x laterality interaction: one such interaction would arise, for example, if in a given M1/S1 cluster there were a systematic greater proportion of peaks derived from a task involving a limb of the same side for the elderly subjects. This analysis showed that this was the case for cluster 45 (right M1/S1), where a higher number of peaks derived from tasks performed with the right side of the body was present for older subjects. This is in line with the so-called “age-related reduction of hemispheric lateralization” at the level of sensorimotor cortices: however, this pattern was independent from the level of performance.

We added this information on page 7:

A further classification of peaks forming the sensorimotor clusters based on the effector/body side is available in Table S3. Based on this last classification, we were able to test the effect called “age-related reduction of

hemispheric lateralization”, whereby older adults might show a greater involvement of sensorimotor regions on the same side of the body involved by the task. We observed that this was true for CL45 (right precentral gyrus/postcentral gyrus): older adults showed a higher number of peaks derived from tasks performed with the right side of the body (#peaks left body side elderly: 5; #peaks left body side young: 4; #peaks right body side elderly: 6; #peaks right body side young: 0; #peaks bilateral body side elderly: 11; #peaks bilateral body side young: 14).

And on page 8:

Moreover, we observed that for CL12 (left postcentral gyrus) older adults showed a similar number of peaks derived from tasks performed with the left -i.e. same- side of the body (#peaks left body side elderly: 4; #peaks left body side young: 5; #peaks right body side elderly: 7; #peaks right body side young: 2; #peaks bilateral body side elderly: 3; #peaks bilateral body side young: 3, Table S3).

Finally, we discussed this on pages 11-12:

On the other hand, we found a reduced recruitment of brain sensorimotor regions, such as the precentral/postcentral gyrus in elderly groups. The peaks forming such clusters derived from nineteen different studies, a mix of motor execution and motor cognitive tasks, listed in Table S2a. **Notably, this cluster was more frequently associated with the use of the contralateral part of the body for young subjects than for the older ones, independently from the level of performance achieved behaviorally. In other words, in older subjects the cluster contained a proportionally greater number of activation peaks associated with ipsilateral movements. This is in line with the so-called “age-related reduction of hemispheric lateralization” at the level of sensorimotor regions already highlighted by Turskey and colleagues (2016) in a meta-analysis conducted on motor execution data⁴¹.**

At line 399, the authors label a finding as premotor, but the finding appears to be in pre/postcentral gyrus, which is generally more considered primary sensorimotor or primary somatomotor cortex. How was the functional label of premotor determined? Care should be taken with these labels. There are studies that help with appropriate functional labeling:

- **Mayka M A, Corcos DM, Leurgans SE, Vaillancourt DE. 2006. Three-dimensional locations and boundaries of motor and premotor cortices as defined by functional brain imaging: a meta-analysis. *Neuroimage*. 31:1453–1474.**

A.R. Thank you for spotting this typo. We corrected the anatomical label in the revised version of the manuscript:

This is what we report here for the first time with a novel meta-analysis on motor control: we found significant interactions in sensorimotor, cerebellar, and occipito-temporal cortices when considering, together with the factor age, the level of task performance achieved, and the specific nature of the task.

At line 421, the characterization of "with increasing cognitive load" is somewhat perplexing. My understanding was that the finding alluded to related to performance, not task complexity.

A.R. Thank you for this comment. We changed “cognitive load” with “task demand”. Indeed, in our paper, we considered the achieved level of behavioral performance as an indirect index of the task demand.

We made it clear on page 3:

In the present paper we address this specific topic by reviewing with a quantitative meta-analytical approach the available task-based brain imaging activation evidence on how the physiological process of aging affects motor control with a variety of experimental tasks. **Crucially, we considered not only explicit motor execution**

tasks, but also cognitive motor tasks (e.g., motor imagery, motor prediction paradigms) and the achieved level of behavioral performance (taken as an indirect index of the task demands). To the best of our knowledge, this is the first meta-analytical attempt to summarize the previous findings on aging in the domain of motor control with this specific approach.

And on page 11:

We classified these experiments based on the specific nature of the task (paradigms involving explicit motor execution or cognitive motor tasks, such as motor imagination/observation paradigms) and on the level of performance achieved by the elderly participants (similar to young subjects or declined, in terms of accuracy or reaction times). We used this last factor as a “proxy” of task demand.

What standards for thresholds were used in identifying foci to include in the meta-analysis. For instance, some studies (maybe more common in older studies) will report on clusters that don't survive correction for multiple comparison.

A.R. The information about the statistical thresholds of studies included in the meta-analysis is now included in Table S1.

It's not clear whether the coordinates from within-group data were treated differently from coordinates from between-group data. Using foci from a cluster that is present in one group but not the other is not the same as using foci from between-group clusters, which the authors seem to acknowledge.

A.R. For all studies that entered in the meta-analysis we used data derived from (i) within group simple effects and (ii) between group comparisons. As in principle, between group differences could arise from a decrease of activity in the reference group, we incorporated also the within group data: this allowed us to have a more complete survey on whether a given brain region was differentially activated across groups, while still being active in each a group above a given conventional threshold, or whether the region, besides being significantly associated with one group, it never reached statistically significant effects in the other group. In any event, for the interaction group-by-task effects we only considered 1st order interactions. This is consistent with several meta-analyses conducted with a similar methodological approach (see for example Paulesu et al., 2014; Devoto et al., 2018, 2020).

We specified this on pages 15-16:

Further, we conducted an up-to-date manual scan of the references of the selected articles, to ensure that all relevant papers had been included. For the suitable studies, we considered data derived from (i) within group simple effects and (ii) between group comparisons. We incorporated also within group data to have a more complete survey on whether a given brain region was differentially activated across groups, while still being active in each group above a given conventional threshold, or whether the region, besides being significantly associated with one group, it never reached statistically significant effects in the other group. The flowchart of the selection process is illustrated in Supplementary Figure 4.

The same groups seem to have multiple studies from around the same time included in the meta-analysis (e.g., Van Impe and Heuninckx). Just to be clear, these do not reflect overlapping groups of participants?

A.R. Accordingly to what the authors reported in their papers, there was no overlapping of participants across the different studies.

We specified this on page 16:

The final dataset comprised 1349 participants, 616 elderly and 607 young participants. Please note that 126 participants cannot be assigned to the young/elderly group since in the original studies participants were not divided in two groups and authors performed a regression analysis on a single group using the variable age as covariate (see^{101,102,109}). **Even if some of the included papers were from the same group of authors^{73,74,78-80,101,102,106-108}, there was no overlapping of participants across the different studies.**

The elderly group age range was 58-80, while the age range for the young group was 21-31.

More minor:

The first part of title is a little strange "motor brain"??

A.R. Following the referee's comment, we changed the title of our manuscript in *How the motor system copes with aging: a quantitative meta-analysis of forty functional imaging studies*.

The abstract alternates between results and interpretations

A.R. We modified the abstract following the referee's suggestion. Please see our revised version of the abstract.

Was the ALE map threshold voxel- or cluster-level corrected?

A.R. We adopted a cluster-level FWE correction. We added this information on page 17.

At line 542, what is "district"?

A.R. Thanks for spotting this typo, we removed it.

There doesn't seem to be a genuine conclusion. the manuscript ends with limitations.

A.R. We added a paragraph with the final remarks of our meta-analysis.

Final remarks

The present study quantitatively reviewed the literature on the aging-related neurofunctional effects on motor control, considering the specific nature of the motor task and the level of performance achieved. Our meta-analysis indicates that motor control is associated with aging-related changes in brain activity, involving not only motoric brain regions but also posterior areas such as the occipito-temporal cortex. Notably, some of these differences depend on the specific nature of the motor task and the level of performance achieved by the participants. These findings support a more general model like the CRUNCH hypothesis, as it better captures the effects of aging in the domain of motor functions, making fewer anatomical assumptions while also considering tasks-dependent and performance-dependent manifestations. Besides the theoretical implications, the present data also provide additional information for the motor rehabilitation domain, indicating that motor control is a more complex phenomenon than previously understood, to which separate cognitive operations can contribute and decrease in different ways with aging.

There are many grammatical and spelling mistakes, and the language at times is difficult to follow; e.g., "none of these models has not taken into great consideration the motor control dimension" or "At variance with what can be done in a novel empirical experiment, in which the variables under examination are controlled by the experimenter, meta-analyses are more observational in nature."

A.R. The English language has been checked throughout the text.

Reviewer #2

In the present paper, Zapparoli et al. performed a quantitative meta-analysis of 40 brain imaging studies to assess the effects of aging on motor control. Results identified higher activation in the occipital cortex and lower activation the sensorimotor cortex for older compared to younger participants, as well as age by performance interactions in sensorimotor, occipito-temporal, and cerebellar regions. Findings were interpreted in line with major theories of neurocognitive aging and found to support mainly the Compensation Related Utilization of Neural Circuits Hypothesis (CRUNCH). This meta-analysis is a needed contribution to an understudied line of investigation. However, a few aspects need to be addressed before the contribution of this paper can be fully assessed, mainly regarding the clarity of the methods and presentation of the results, and a discussion section that seems somewhat unfocused and overly speculative.

A.R. We would like to thank the reviewer for her/his overall positive evaluation of our manuscript.

It is not entirely clear what the advantage of the current method is, compared to the widely-used ALE procedure, and why the authors preferred this approach if they ended up “masking” with ALE results? Please clarify.

A.R. The CluB toolbox permits to extract a set of spatially coherent clusters from a database of stereotactic coordinates and to explore each single cluster of activation for its composition according to the cognitive dimensions of interest. This last step, called “cluster composition analysis,” permits to explore neurocognitive effects by adopting a factorial-design logic and by testing the working hypotheses using either asymptotic tests or exact tests. This is something that cannot be done with GingerALE. Indeed, meta-analyses based on ALE are limited by the need of testing regional functional anatomical effects from highly homogeneous studies permitting, at the most, the evaluation of the neurofunctional differences (e.g., Group A > Group B) and commonalities (e.g., overlap of the effects of Group A & Group B) between two classes of studies. Thus, ALE cannot test more complex factorial models, the level of analysis needed for a complex neurocognitive scenario like the one behind aging and the nature of the task/level of the performance achieved.

In turn, the hierarchical clustering procedure does not provide a statistical test of the spatial significance of the resulting clusters; this can be compensated for by searching for spatial convergence between the clustering solution of CluB and the results of an Activation Likelihood Estimate (ALE)-based meta-analysis on the same overall dataset submitted to a correction for multiple comparisons. For the spatial cross-validation with ALE we employed the Turkeltaub Non-Additive method (Turkeltaub et al. 2012), with the general statistical threshold set to $p < 0.05$ FWE corrected. The resulting maps were overlapped with the HC map with the “intersection” function in the software MRICron ([https:// www.nitrc.org/projects/mricron](https://www.nitrc.org/projects/mricron)). Only the clusters that fell in this intersection map were then considered for further analyses (the cluster composition analysis) and discussion.

We specify this on page 16 of the revised manuscript:

To identify anatomically coherent regional effects, we first performed a hierarchical clustering analysis (HC) using the unique-solution clustering algorithm developed by Cattinelli et al. (2013). This method, implemented in a suite of MATLAB (2014a MathWorks) and C++ scripts called CluB (Clustering the Brain, Berlinger et al. 2019).

The CluB toolbox permits both to extract a set of spatially coherent clusters of activations from a database of stereotactic coordinates, and to explore each single cluster of activation for its composition according to the cognitive dimensions of interest. This last step, called “cluster composition analysis,” permits to explore neurocognitive effects by adopting a factorial-design logic and by testing the working hypotheses using either

asymptotic tests, or exact tests either in a classic inference, or in a Bayesian-like context. This is something that cannot be done with GingerALE. Indeed, meta-analyses based on ALE are limited by the need of testing regional functional anatomical effects from highly homogeneous studies permitting, at the most, the evaluation of the neurofunctional differences (e.g., Group A > Group B) and commonalities (e.g., conjunction effect of Group A & Group B) between two classes of studies. The software cannot test more complex factorial models, the level of analysis needed for a complex neurocognitive scenario like the one behind aging and the nature of the task/level of the performance achieved.

Specifically, the CluB toolbox considers the squared Euclidean distance between each pair of foci included in the dataset. The clusters with minimal dissimilarity are recursively merged using Ward's criterion (Ward 1963), to minimize the intra-cluster variability and maximizing the between-cluster sum of squares (Cattinelli et al. 2013). To impose a suitable a priori spatial resolution to our analyses, we set this to be 7 mm in terms of the maximum mean spatial variance within each cluster in the three directions. The centroid coordinates of each resulting cluster were then labeled according to the Automatic Anatomic Labelling (AAL) and then controlled by visual inspection on the MRIcron (Rorden and Brett 2000) visualization software.

Moreover, on page 17, we wrote:

As the HC procedure does not provide a statistical test of the spatial significance of the resulting clusters, this can be compensated for by searching for spatial convergence between the clustering solution and the results of an Activation Likelihood Estimate (ALE)-based meta-analysis on the same overall dataset (see, for example (Fisher 1970, Paulesu, Danelli and Berlinger 2014)). For the spatial cross-validation with ALE we employed the Turkeltaub Non-Additive method (Turkeltaub et al. 2012), with the general statistical threshold set to $p < 0.05$ FWE corrected. The resulting maps were overlapped with the HC map with the “intersection” function in the software MRIcron ([https:// www.nitrc.org/projects/mricron](https://www.nitrc.org/projects/mricron)). Only the clusters that fell in this intersection map were then considered for further analyses (the cluster composition analysis) and discussion.

In the methods section, for the 3-way interaction, it is stated that it was not “associated with a formal statistical test”. Yet, in the results, the authors talk about a group x task x performance interaction.

A.R. The referee is right: a three-way interaction cannot be formally tested. However, we observed that the cluster located in the right occipito-temporal cortex showed a significant interaction Group*Performance and Performance*Task interactions, suggesting a possible three-way interaction. Thus, we plotted the distribution of the different peaks along the different levels/factors to show that this cluster is more frequently activated by the elderly subjects, during cognitive motor tasks, when the level of performance is similar to the younger counterparts.

Accordingly, we amended the manuscript highlighting that this is a qualitative exploration of the peaks' distribution and not a formal statistical assessment of a 3-way interaction.

On pages 9-10 we wrote:

Group-by-Task-by-Performance Interaction qualitative exploration

Two clusters, located in the left sensorimotor cortices (CL12) and in the right occipito-temporal region (CL66), displayed both a significant group-by-performance and performance-by-task interaction effect (Table 2, Figures 4a and 4b), suggesting a possible three-way interaction effect.

The inspection of the graph in Fig. 4a shows that the left sensorimotor region was more likely to be associated with the elderly group during the execution of motor execution tasks, when their performance is similar to their young counterparts and with the young participants group during the execution of cognitive motor tasks, when they achieved a better behavioral performance.

Precentral / Postcentral Gyrus (CL12)

X=-53, Y=-6, Z=-35

Figure 4a. Clusters showing a significant Group-by-Performance and a Task-by-Performance interaction effect.

The inspection of the graph in Fig. 4b shows that this region was more likely to be associated with the elderly group during the execution of cognitive motor tasks, when their performance is similar to their young counterparts. Interestingly, this cluster falls within the boundaries of a right extra-striate body area⁴⁵ and its detection was associated with cognitive motor tasks including action observation tasks and motor imagery for body parts^{46,47}.

Occipito-temporal region (CL66)

X=47, Y=-66, Z=-2

Figure 4b. Clusters showing a significant Group-by-Performance and a Task-by-Performance interaction effect.

It is unclear what the “undifferentiated clusters” are. For instance, CL39 (mentioned in the text) does not show any significant effects in Table 2. Please clarify. Also, please check the correspondence between main text and Table 2.

A.R. Following the referee’s comment, we decided to do not report these “undifferentiated clusters”. These were spatially significant clusters, consistently activated by motor tasks, yet with no association with elderly or young groups (i.e., the p-value of the binomial test executed over the levels of the Group factor being greater than 0.5).

In its present form, the discussion seems to be overly long and lacking focus, especially the “Neurophysiological aging effects” part. I think more emphasis should be given to how the present findings specifically provide support for or against the theories of neurocognitive aging mentioned in the introduction.

A.R. Thanks for this comment. We shortened the paragraph on the neurophysiological effects of aging, and we gave more emphasis to the last paragraph.

Please see our revised discussion on pages 11-12.

Minor comments:

- **I found Table 2 difficult to read, mainly due to its width. I would suggest listing the coordinates for both hemispheres in the same column, preceded by a column with L/R entries.**

A.R. We modified Table 2 accordingly.

The text needs careful proofreading as there are a number of typos and awkward phrases that affect comprehension.

A.R. The English language has been checked throughout the text.

Reviewers' comments:

Reviewer #1 (Remarks to the Author):

The original submission of the manuscript generated a multitude of concerns. The authors have put forth a commendable effort addressing these concerns and bolstering their manuscript, but I still have several comments:

The "review" that I cited should have been:

Seidler et al. (2010) "Motor control and aging: Links to age-related brain structural, functional, and biochemical effects" *Neuroscience and Biobehavioral Reviews* 34: 721-733.

rather than the empirical study:

Bernard JA, Seidler RD. 2012. Evidence for motor cortex dedifferentiation in older adults. *Neurobiol Aging*. 33:1890-1899.

My apologies for this confusion. Regardless, de-differentiation (and the central debate in the aging motor system domain, i.e., between compensation and de-differentiation) is not mentioned until the discussion. Further, I found it difficult keeping track of and relating the findings to all the different, partially overlapping theories (HAROLD, PASA, CRUNCH, compensation, de-differentiation, maybe others?). I recognize that I am somewhat to blame for highlighting de-differentiation, but this is central to the discussion on the aging motor system, and therefore seemed necessary to discuss.

The specifications for novelty were helpful, but the present study is still less impactful given the prior work on the aging motor system.

"specific nature of the task" is a little vague. Is there a clearer way to describe motor execution versus cognitive motor?

it might behoove the authors to double-check on de-differentiation--i'm not sure it stipulates a negative correlation between activation and performance. no association may also be sufficient, as long as it isn't the positive correlation as for compensation.

not critical, but might help to know whether if the laterality effects mentioned (distribution of #s of clusters) were significant?

the Table S1 heading indicates that sample size should be included in the table, but I didn't see it there

the "Effect" column in table S3 is difficult to understand. E.g., to what is "Group effect (Young), Task effect (Motor execution task), Interaction effect (Performance- by-task)" referring?

Addressing the issue of SM1 somatotopy, while I commend the authors for generating a new table with specific body parts underlying clusters, I'm not sure that the explanation provided addresses the

original concern: i.e., if there are a number of different studies involving movement of different body parts, then the activation foci along SM1 from these constituent studies would be spread too far apart to converge in a meta-analysis. This would limit the ability to test age-related bilateralization because there would not be much convergence in SM1 (false negatives). The concern was not about resolution but about distantly located foci that would not converge.

However, one argument against this is that most of the constituent studies employed hand movement tasks, which confines foci to a limited spot along SM1. In addition, maybe the authors are more interested in identifying which brain regions exhibit differential activation according to group, group x task, etc., independent of part of body used (as they alluded to).

Reviewer #2 (Remarks to the Author):

The authors have addressed my comments satisfactorily and the manuscript has substantially gained in clarity. There are still some remaining typos that should be double-checked. Other than that, I have no other comments.

Reviewers' comments:

Reviewer #1 (Remarks to the Author):

The original submission of the manuscript generated a multitude of concerns. The authors have put forth a commendable effort addressing these concerns and bolstering their manuscript, but I still have several comments:

A.R. We thank the referee for her/his appreciation of our revision work.

The "review" that I cited should have been:

Seidler et al. (2010) "Motor control and aging: Links to age-related brain structural, functional, and biochemical effects" *Neuroscience and Biobehavioral Reviews* 34: 721-733.

rather than the empirical study:

Bernard JA, Seidler RD. 2012. Evidence for motor cortex dedifferentiation in older adults. *Neurobiol Aging*. 33:1890–1899.

My apologies for this confusion. Regardless, de-differentiation (and the central debate in the aging motor system domain, i.e., between compensation and de-differentiation) is not mentioned until the discussion.

A.R. Thank for bringing our attention to this interesting review, which is now cited in the introduction section. Moreover, as the referee correctly suggested, we introduce the de-differentiation hypothesis in the paragraph describing the different neurocognitive theories on aging.

On page 5, we wrote:

Aging-related prefrontal hyperactivations are also reported by studies addressing the effects of aging on motor control. Indeed, Seidler and collaborators (2010) reviewed this literature to report that older adults show additional activations in higher-level prefrontal region, but also in sensorimotor cortical areas. Notably, the activity of these prefrontal and sensorimotor regions was often associated with better performance, suggesting the possibility that the engagement of additional areas may represent a compensatory process taking place during motor task performance (Seidler et al., 2010).

On pages 3-4, we wrote:

On the one hand, the **compensation hypothesis** predicts that age-related increases in brain activation, as well as the recruitment of additional areas, compensate for various neural/behavioural deficits^{15,29,62-64}. According to the compensation hypothesis, the **overactivations would be larger in good than in poor performers**. Compensatory patterns have been documented for several cognitive domains (e.g., working memory, episodic memory retrieval, perception, inhibitory control, etc,^{15,29,62-64}). This phenomenon was initially observed in the prefrontal cortex; moreover, compensatory processes have been described as reduced inter-hemispheric

asymmetries for tasks associated with strongly lateralized fMRI patterns in younger participants²⁹. This overall pattern is well captured by the *HAROLD model*⁶².

A similar compensatory hypothesis is covered by the *PASA model*³⁰, whereby aging is associated with a significant increment of the frontal lobes' activation and reduced neural activity in posterior areas (mainly occipito-temporal cortices). Age-related reductions in occipital activity have been attributed to deficits in sensory processing, while age-related increased prefrontal activity would represent an attempt to compensate for these deficits.

On the other hand, the **dedifferentiation hypothesis** posits that age-related brain functional changes might indicate a generalized diffusion of brain activity attributable to deficits in neurotransmission, which in turn causes a loss of neural specialization. This hypothesis suggests that hyperactivations should not be accompanied by successful behavioural performance⁶⁷. Several experiments have previously supported the dedifferentiation hypothesis. For example, Loibl et al.⁶⁶ observed age-related increased activations in ipsilateral motor areas (M1 and SMA), that were negatively correlated with motor performance⁶⁶. Similar results were found by Bernard et al.⁶⁷ with transcranial magnetic stimulation, reporting for the elderly subjects more diffuse motor cortical excitability in the hemisphere contralateral and ipsilateral to the recorded motor evoked potentials. Notably, this broader excitability correlated with augmented reaction times⁶⁷.

Further, I found it difficult keeping track of and relating the findings to all the different, partially overlapping theories (HAROLD, PASA, CRUNCH, compensation, de-differentiation, maybe others?). I recognize that I am somewhat to blame for highlighting de-differentiation, but this is central to the discussion on the aging motor system, and therefore seemed necessary to discuss.

A.R. We agree with the referee and we re-shaped our discussion, with clear indication about how our findings relate to the different yet partially overlapping neurocognitive theories.

On page 12, we wrote:

Taken together, our significant group effects indicating a more frequent recruitment of occipital regions in elderly subjects during the execution of a variety of motor tasks are in contrast with the hypotheses made by the *HAROLD* (Hemispheric Asymmetry Reductions in Older Adults) and the *PASA* pattern (Posterior-Anterior Shift in Aging) models, whereby aging should be accompanied by augmented and bilateral recruitment of the frontal lobes and reduced posterior activations. On the contrary, we found reduced brain activations only at the level of sensorimotor cortices.

On pages 12-13, we wrote:

It is important to emphasize that, beyond mere group effects, we also found significant interactions between factor age, the level of task performance achieved, and the specific nature of the task with respect to the presence of an overt motor output: these were localized in sensorimotor and occipito-temporal cortices and in the cerebellum. The functional meaning of these interactions varied in the different regions. On the one hand, the interaction in the **occipito-temporal cluster** can be interpreted as a **successful compensation**. In this cluster, brain activity was higher when elderly participants achieved a level of performance similar to their younger counterparts, while the number of peaks associated with a deteriorated performance was significantly lower.

On the other hand, the interaction in sensorimotor regions can be differently interpreted depending on the performance achieved by the older adults: the peaks forming this cluster derived from studies characterized by a declined behaviour, but also from experiments in which the elderly group reach the same performance level of younger subjects. Accordingly, there are indications that the activity in this region can reflect both successful and failed compensatory processes. This latter scenario is explained by the fact that this region is also significantly hypo-activated in the elderly subjects in the presence of a declined performance.

Finally, the interaction at the level of the cerebellum can be seen as an example of dedifferentiation: indeed, the higher number of peaks associated with the elderly subjects comes from studies where their performance was worse than the younger ones. This would be in line with what observed in previous studies: for example, Carp and colleagues (2011) reported how motor distinctiveness, defined using multivariate pattern analysis, was reduced in older adults in a series of brain regions, including the cerebellum⁶⁸. However, there is one conceptual difficulty when considering the cerebellum in the context of a dedifferentiation hypothesis, given the motoric nature of the tasks behind this result.

Indeed, the cerebellum contributes to motor performance in a dynamic manner, with a greater contribution during the learning phase of a new motor skill (see, for example⁶⁹). Accordingly, it is conceivable that the more frequent activation of the cerebellum in older adults (when they showed a declined performance) might be due to the fact that they perceived motor tasks as novel and less automatized, as when new motor learning is occurring or as a sign of a lost motor automaticity.

These results are in line with what hypothesized by the CRUNCH hypothesis. Indeed, we showed that at relatively low levels of task demand/good performance, one can observe region-specific hyperactivations in older subjects (right occipito-temporal and left sensorimotor clusters). Interestingly, such compensatory overactivations depended on the specific nature of the experimental task, with sensorimotor areas associated with motor execution tasks and posterior regions recruited for cognitive motor tasks (see the exploration of the three-way interaction). With the increase of the task demand/decrease of performance, the attempt at compensation becomes less successful, and it can be defined as an unsuccessful attempt or an expression of dedifferentiation processes (right cerebellar cluster). Beyond a certain level of task demand, the elderly brain does not show sufficient activation levels, hence performance declines relative to the younger group (left sensorimotor cluster). Overall, this pattern supports the idea that the CRUNCH and the dedifferentiation hypotheses might not be incompatible (see for example⁶⁸).

The specifications for novelty were helpful, but the present study is still less impactful given the prior work on the aging motor system.

A.R. As much as we respect the referee's opinion, we maintain our position. We believe that our paper represents a substantially novel contribution to the literature: this is the first meta-analysis on aging and motor control that considered not only the aging factor, but also the nature of the task (with reference to the presence/absence of an overt motor output), the level of performance (equal or not equal to the younger counterparts, performance being used as a proxy of task demands) and their interactions and how these impact on brain activity. Crucially, our meta-analytical approach allowed us to explore these effects by adopting a factorial design - not affordable using the ALE technique - and by testing the working hypotheses using either asymptotic tests or exact tests.

This is something never done before.

"specific nature of the task" is a little vague. Is there a clearer way to describe motor execution versus cognitive motor?

A.R. We now specify that the factor "specific nature of the task" refers to the presence/absence of an overt motor output (see for example, page 5).

it might behove the authors to double-check on de-differentiation--i'm not sure it stipulates a negative correlation between activation and performance. no association may also be sufficient, as long as it isn't the positive correlation as for compensation.

A.R. We amended the text accordingly on page 4:

On the other hand, the **dedifferentiation hypothesis** posits that age-related brain functional changes might indicate a generalized diffusion of brain activity attributable to deficits in neurotransmission, which in turn causes a loss of neural specialization. This hypothesis suggests that hyperactivations should not be accompanied by successful behavioural performance⁶⁷. Several experiments have previously supported the dedifferentiation hypothesis. For example, Loibl et al.⁶⁶ observed age-related increased activations in ipsilateral motor areas (M1 and SMA), that were negatively correlated with motor performance⁶⁶. Similar results were found by Bernard et al.⁶⁷ with transcranial magnetic stimulation, reporting for the elderly subjects more diffuse motor cortical excitability in the hemisphere contralateral and ipsilateral to the recorded motor evoked potentials. Notably, this broader excitability correlated with augmented reaction times⁶⁷.

not critical, but might help to know whether if the laterality effects mentioned (distribution of #s of clusters) were significant?

A.R. Thanks for this comment. To test this hypothesis, we performed a chi-squared analysis (Factor1, Group: Elderly/Young; Factor2, body side: left/right). The chi-squared test of independence showed that there was a trend of significance for CL45 (right precentral gyrus/postcentral gyrus, $X^2(1, N=15)=3.6$, $p=0.056$), but not for CL12 (left postcentral gyrus, $X^2(1, N=18)=2.1$, $p=0.15$).

We added this on page 7:

Based on this last classification, we were able to test the effect called “age-related reduction of hemispheric lateralization”, whereby older adults might show a greater involvement of sensorimotor regions on the same side of the body involved by the task. We observed that this was true for CL45 (right precentral gyrus/postcentral gyrus). Indeed, the chi-squared analysis (Factor1, Group: Elderly/Young; Factor2, body side: left/right) showed a trend of significance ($X^2(1, N=15)=3.6$, $p=0.056$): older adults showed a higher number of peaks derived from tasks performed with the right side of the body (#peaks left body side elderly: 5; #peaks left body side young: 4; #peaks right body side elderly: 6; #peaks right body side young: 0; #peaks bilateral body side elderly: 11; #peaks bilateral body side young: 14).

And on page 8:

Moreover, we observed that for CL12 (left postcentral gyrus) older adults showed a similar number of peaks derived from tasks performed with the left -i.e. same- side of the body ($X^2(1, N=18)=2.1$, $p=0.15$; #peaks left body side elderly: 4; #peaks left body side young: 5; #peaks right body side elderly: 7; #peaks right body side young: 2; #peaks bilateral body side elderly: 3; #peaks bilateral body side young: 3, Table S3).

the Table S1 heading indicates that sample size should be included in the table, but I didn't see it there

A.R. The referee is right. We removed the column of the sample size since it was already present in Table 1. We corrected the caption of Table S1.

the "Effect" column in table S3 is difficult to understand. E.g., to what is "Group effect (Young), Task effect (Motor execution task), Interaction effect (Performance- by-task)" referring?

A.R. It refers to the statistical effects generating that specific clusters. We modified the caption of Table S3.

Table S3 | Details of the peaks forming M1/S1 clusters showing a significant Group Effect, Group-by-Performance and Group-by-Task effects.

Addressing the issue of SM1 somatotopy, while I commend the authors for generating a new table with specific body parts underlying clusters, I'm not sure that the explanation provided addresses the original concern: i.e., if there are a number of different studies involving movement of different body parts, then the activation foci along SM1 from these constituent studies would be spread too far apart to converge in a meta-analysis. This would limit the ability to test age-related bilateralization because there would not be much convergence in SM1 (false negatives). The concern was not about resolution but about distantly located foci that would not converge.

However, one argument against this is that most of the constituent studies employed hand movement tasks, which confines foci to a limited spot along SM1. In addition, maybe the authors are more interested in identifying which brain regions exhibit differential activation according to group, group x task, etc., independent of part of body used (as they alluded to).

A.R. We mentioned in the introduction section that our approach was aimed at investigating whether potential aging-related neurofunctional effects might occur regardless of the specific effector and/or the side used to execute the task.

Nevertheless, we discuss this a potential limitation on page 13:

Moreover, we should recognize that our approach cannot entirely test the HAROLD model and its adaptation to motor control (the so-called “sensorimotor lateralization hypothesis”) since we included experimental tasks executed with both body sides and different effectors. This was done to investigate whether potential aging-related neurofunctional effects might occur regardless of the specific effector and/or the side used to execute the task. However, it should be acknowledged that most of the constituent studies employed hand movement tasks, which confines foci to a limited spot along the sensorimotor cortices.

Reviewer #2 (Remarks to the Author):

The authors have addressed my comments satisfactorily and the manuscript has substantially gained in clarity. There are still some remaining typos that should be double-checked. Other than that, I have no other comments.

A.R. We thank the referee for her/his appreciation of our revision work. The English language has been checked again throughout the text.